# Preference Optimization for Combinatorial Optimization Problems

**Mingjun Pan** [* 1 2]  **Guanquan Lin** [* 3]  **You-Wei Luo** [4 5]  **Bin Zhu** [2]  **Zhien Dai** [6]  **Lijun Sun** [1]  **Chun Yuan** [3]

## Abstract

Reinforcement Learning (RL) has emerged as a powerful tool for neural combinatorial optimization, enabling models to learn heuristics that solve complex problems without requiring expert knowledge. Despite significant progress, existing RL approaches face challenges such as diminishing reward signals and inefficient exploration in vast combinatorial action spaces, leading to inefficiency. In this paper, we propose ***Preference Optimization***, a novel method that transforms quantitative reward signals into qualitative preference signals via statistical comparison modeling, emphasizing the superiority among sampled solutions. Methodologically, by reparameterizing the reward function in terms of policy and utilizing preference models, we formulate an entropy-regularized RL objective that aligns the policy directly with preferences while avoiding intractable computations. Furthermore, we integrate local search techniques into the fine-tuning rather than post-processing to generate high-quality preference pairs, helping the policy escape local optima. Empirical results on various benchmarks, such as the Traveling Salesman Problem (TSP), the Capacitated Vehicle Routing Problem (CVRP) and the Flexible Flow Shop Problem (FFSP), demonstrate that our method significantly outperforms existing RL algorithms, achieving superior convergence efficiency and solution quality.

## 1. Introduction

Combinatorial Optimization Problems (COPs) are fundamental in numerous practical applications, including route planning, circuit design, scheduling, and bioinformatics (Papadimitriou & Steiglitz, 1998; Cook et al., 1994; Korte et al., 2011). These problems require finding an optimal solution from a finite but exponentially large set of possibilities and have been extensively studied in the operations research community. While computing the exact solution is impeded by their NP-hard complexity (Garey & Johnson, 1979), efficiently obtaining near-optimal solutions is essential from a practical standpoint.

Deep learning, encompassing supervised learning and reinforcement learning, has shown great potential in tackling COPs by learning heuristics directly from data (Bengio et al., 2021; Vinyals et al., 2015). However, supervised learning approaches heavily rely on high-quality solutions, and due to the NP-hardness of COPs, such collected training datasets may not guarantee optimality, which can lead models to fit suboptimal policies. In contrast, RL has emerged as a promising alternative, achieving success in areas involving COPs such as mathematical reasoning (Silver et al., 2018) and chip design (Mirhoseini et al., 2021). (Deep) RL-based solvers leverage neural networks to approximate policies and interactively obtain rewards/feedback from the environment, allowing models to improve in a trial-and-error manner. (Bello et al., 2016; Kool et al., 2019).

Despite its potential, applying RL to COPs presents significant challenges. **Diminishing reward signals**: As the policy improves, the magnitude of advantage value decreases significantly. Since RL rely on these numerical signals to drive learning, the reduction in their scale leads to vanishing gradients and slow convergence. **Unconstrained action spaces**: The vast combinatorial action spaces complicate efficient exploration, rendering traditional exploration techniques like entropy regularization of trajectories computationally infeasible. **Additional inference time**: While neural solvers are efficient in inference, they often suffer from finding near-optimal solutions. Many works adopt techniques like local search as a post-processing step to further improve solutions, but this incurs additional inference costs.

To address the issue of diminishing reward signals and inefficient exploration, we propose transforming quantitative reward signals into qualitative preference signals, focusing on the superiority among generated solutions rather than their absolute reward values. This approach stabilizes the

---
[*]Equal contribution  [1]China Mobile [2]Peking University [3]Tsinghua University [4]Jiaying University [5]Sun Yat-Sen University [6]Central South University
Mingjun Pan <mingjunpan96@gmail.com>. Correspondence to: You-Wei Luo <luoyw28@mail2.sysu.edu.cn>, Chun Yuan <yuanc@sz.tsinghua.edu.cn>.

*Proceedings of the 42nd International Conference on Machine Learning*, Vancouver, Canada. PMLR 267, 2025. Copyright 2025 by the author(s).

learning process and theoretically emphasizes optimality, as preference signals are insensitive to the scale of rewards. By deriving our method from an entropy-regularized objective, we inherently promote efficient exploration within the vast action space of COPs, overcoming the computational intractability associated with traditional entropy regularization techniques. Additionally, to mitigate the extra inference time induced by local search, we integrate such techniques into the fine-tuning process rather than using them for post-processing, which enables the policy to learn from improved solutions without incurring additional inference time.

Furthermore, preference-based optimization has recently gained prominence through its application in Reinforcement Learning from Human Feedback (RLHF) for large language models (Christiano et al., 2017; Rafailov et al., 2024; Meng et al., 2024). Inspired by these advancements, we introduce a novel update scheme that bridges preference-based optimization with COPs, leading to a more effective and consistent learning process. In this work, we propose a novel algorithm named Preference Optimization (PO), which can seamlessly substitute conventional policy gradient methods in many contexts. In summary, our contributions are:

1. **Preference-based Framework for RL4CO:** We present a novel method that transforms quantitative reward signals into qualitative preference signals, ensuring robust learning independent of reward scaling. This addresses the diminishing reward differences common in COPs, stabilizing training and consistently emphasizing better solutions with relation preservation.

2. **Reparameterized Entropy-Regularized Objective:** By reparameterizing the reward function in terms of policy and leveraging statistical preference models, we formulate an entropy-regularized objective that aligns the policy directly with preferences. This approach bypass the intractable enumeration of the entire action space, maintaining computational feasibility.

3. **Integration with Local Search:** We demonstrate how our framework naturally incorporates heuristic local search for fine-tuning, rather than relegating it to post-processing. This integration aids trained solvers in escaping local optima and enhances solution quality without introducing additional time at inference.

Extensive experiments across a diverse range of COPs validate the efficiency of our proposed PO framework, which achieves significant acceleration in convergence and superior solution quality compared to existing RL algorithms.

## 2. Related Work

**RL-based Neural Solvers.** The pioneering application of Reinforcement Learning for Combinatorial Optimization problems (RL4CO) by (Bello et al., 2016; Nazari et al., 2018; Kool et al., 2019) has prompted subsequent exploration on frameworks and paradigms. We classify the majority of RL4CO research from the following perspectives:

*End-to-End Neural Solvers.* Numerous works have focused on designing end-to-end neural solvers that directly map problem instances to solutions. Techniques exploiting the inherent equivalence and symmetry properties of COPs have been introduced to facilitate near-optimal solutions (Kwon et al., 2020; Kim et al., 2022; Ouyang et al., 2021; Kim et al., 2023). For example, POMO (Kwon et al., 2020) employs multiple diverse starting points to improve training efficiency, while Sym-NCO (Kim et al., 2022) leverages problem symmetries to boost performance. Other studies incorporate entropy regularization at the step level to encourage exploration and improve solution diversity (Xin et al., 2021a; Sultana et al., 2020). Further efforts to enhance generalization include diversifying training datasets to handle a broader range of problem instances (Bi et al., 2022; Wang et al., 2024; Zhou et al., 2023; Jiang et al., 2024). Although most of these works primarily focus on architectural or learning-paradigm improvements, less efforts has been paid to developing novel optimization framework.

*Hybrid Solvers.* Another promising direction integrates neural methods with conventional optimization techniques, combining established heuristics such as $k$-Opt, Ant Colony Optimization, Monte Carlo Tree Search, or Lin-Kernighan to refine solution quality (d O Costa et al., 2020; Wu et al., 2021; Ye et al., 2023; Xin et al., 2021b). For instance, NeuRewriter (d O Costa et al., 2020) couples deep learning with graph rewriting heuristics, while NeuroLKH (Xin et al., 2021b) embeds neural methods into the LKH algorithm. In many cases, these heuristics function as post-processing steps to refine near-optimal solutions (Fu et al., 2021; Ma et al., 2021; Ouyang et al., 2021), but the associated additional inference time can reduce efficiency and may be infeasible for time-critical scenarios.

In this work, we primarily focus on end-to-end neural solvers because hybrid methods rely heavily on heuristic-based solution generation, making it difficult to evaluate the algorithmic impact of RL-based approaches. Therefore, considering the end-to-end modeling can provide a practical evaluation of how different algorithms affect performance.

**Preference-based Reinforcement Learning.** Preference-based reinforcement learning (PbRL) is another area related to our work, which has been widely studied in offline RL settings. PbRL involves approximate the ground truth reward function from preference information rather than relying on explicit reward signals (Wirth et al., 2017). This approach is particularly useful when reward signals are sparse or difficult to specify. Recently, works such as (Hejna & Sadigh, 2024; Rafailov et al., 2024; Meng et al.,

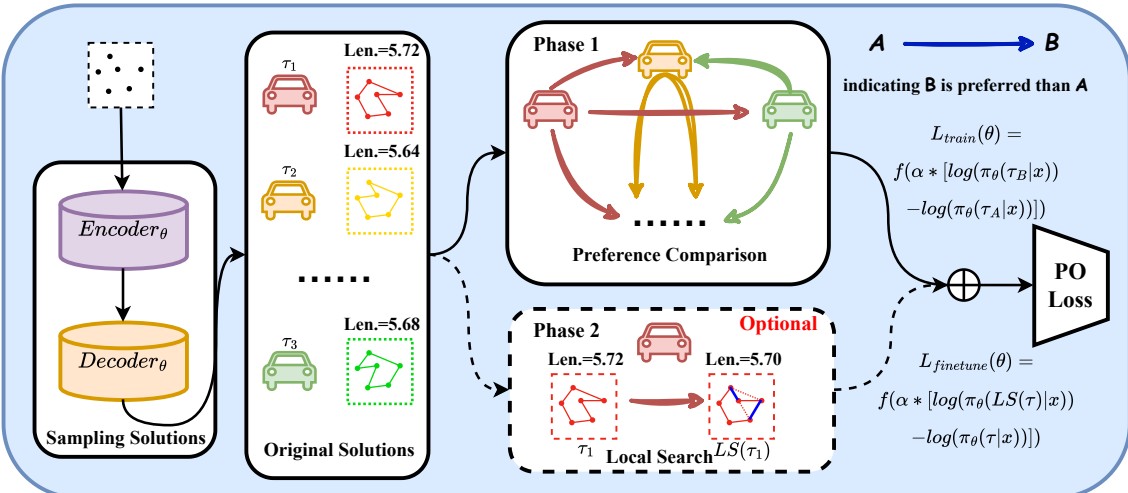

Figure 1: Algorithmic framework of PO for COPs. In the Preference Comparison module, pairwise comparisons are conducted between solutions based on their grounding quality (e.g., trajectory length). The Local Search slightly refines solution $\tau$ to produce improved solution $LS(\tau)$ which contribute additional preference signals $L_{\text{finetune}}$ during fine-tuning.

2024) have proposed novel paradigms to directly improve the KL-regularized policy without the need for learning an approximate reward function, leading to more stable and efficient training. This has led to the development of a series of works (Azar et al., 2024; Park et al., 2024; Hong et al., 2024) in the RLHF phase within language-based models, where preference information is leveraged to align large language models effectively.

Our work bridges the gap between these domains by introducing a preference-based optimization framework specifically tailored for COPs. By transforming quantitative reward signals into qualitative preferences, we address key challenges in RL4CO, such as diminishing reward differences and exploration inefficiency, while avoiding the need for explicit reward function approximation as in PbRL.

## 3. Methodology

In this section, we first recap Reinforcement Learning for Combinatorial Optimization (RL4CO), and Preference-based Reinforcement Learning (PbRL). Next, we explain how to leverage these techniques to develop a novel framework to efficiently train neural solvers that rely solely on relative superiority among generated solutions. Subsequently, we investigate the compatibility of our approach with Local Search techniques for solver training, i.e. fine-tuning. The algorithmic framework of our method is illustrated in Fig. 1.

### 3.1. Reinforcement Learning for Combinatorial Optimization Problems

RL trains an agent to maximize cumulative rewards by interacting with an environment and receiving numerical reward signals. In COPs, the state transitions are typically modeled as deterministic. A commonly used method is REINFORCE (Sutton & Barto, 2018) and its variants, whose update rule is given by:

$$\nabla_\theta J(\theta) = \mathbb{E}_{x \sim \mathcal{D}, \tau \sim \pi_\theta(\tau|x)} \left[ (r(x, \tau) - b(x)) \nabla_\theta \log \pi_\theta(\tau \mid x) \right]$$
$$\approx \frac{1}{|\mathcal{D}|} \sum_{x \in \mathcal{D}} \frac{1}{|S_x|} \sum_{\tau \in S_x} \left[ (r(x, \tau) - b(x)) \nabla_\theta \log \pi_\theta(\tau \mid x) \right], \quad (1)$$

where $\mathcal{D}$ is the dataset of problem instances, $x \in \mathcal{D}$ represents an instance, $S_x$ is the set of sampled solutions (trajectories) for $x$, $r(x, \tau)$ is the reward function derived from distinct COPs and $b(x)$ represents the baseline used to calculate the advantage function $A(x, \tau) = r(x, \tau) - b(x)$, which helps reduce the variance of the gradient estimator.

A critical challenge in REINFORCE-based algorithms is their sensitivity to baseline selection. As demonstrated in (Kool et al., 2019), training without an appropriate baseline led to significantly degraded performance in COP applications. The policy $\pi_\theta(\tau \mid x)$ defines a distribution over trajectories $\tau = (a_1, a_2, \ldots, a_T)$ given the instance $x$. Each trajectory $\tau$ is a sequence of actions generated by the policy: $\pi_\theta(\tau \mid x) = \prod_{t=1}^T \pi_\theta(a_t \mid s_{t-1})$, with $s_0$ being the initial state determined by $x$, and $s_t$ representing the state at time step $t$, which depends on the previous state and the taken action (e.g., $s_t = \mathcal{T}(s_{t-1}, a_t)$). Here, $\mathcal{T}(\cdot)$ represents the state transition function, which is deterministic in COPs. The action $a_t$ is selected by the policy based on state $s_{t-1}$.

Unlike popular RL environments such as Atari (Bellemare et al., 2013) and Mujoco (Todorov et al., 2012), which provide stable learning signals, COPs present distinctive challenges. As the policy improves, the magnitude of advantage value diminishes significantly. Specifically,

$|r(x, \tau) - b(x)| < \epsilon$, where $\epsilon$ is small except for the initial training process. This leads to negligible updates to the policy objective $J(\theta)$ in REINFORCE-variants , which heavily relies on the advantage value $A(x, \tau) = r(x, \tau) - b(x)$. Consequently, the policy struggles to escape local optima during later training stages.

### 3.2. Preference-based Reinforcement Learning

Preference-Based Reinforcement Learning (PbRL) (Wirth et al., 2017) trains an agent using preference labels instead of explicit reward signals obtained through direct interaction with the environment. Concretely, we assume access to a preference dataset $\mathcal{D}_p = \{(\tau_1, \tau_2, y)\}$, where each triplet consists of two trajectories $\tau_1, \tau_2$ and a preference label $y \in \{0, 1\}$. A label $y = 1$ indicates that $\tau_1$ is preferred over $\tau_2$ ($\tau_1 \succ \tau_2$), while $y = 0$ indicates the opposite.

These preferences are assumed to be generated by an underlying (latent) reward function $\hat{r}(x, \tau)$. Various models, such as Bradley-Terry (BT), Thurstone (David, 1963), and Plackett-Luce (PL) (Plackett, 1975), link differences in reward values to observed preferences, enabling us to formulate an objective for learning the reward function.

In paired preference models like BT and Thurstone, a function $f(\cdot)$ maps the difference in rewards to a probability of one trajectory being preferred over another:

$$p^*(\tau_1 \succ \tau_2) = f\left(\hat{r}(x, \tau_1) - \hat{r}(x, \tau_2)\right), \qquad (2)$$

where the Bradley-Terry model adopts the sigmoid function $\sigma(x) = (1 + e^{-x})^{-1}$, and the Thurstone model uses the CDF $\Phi(x)$ of the normal distribution.

This relationship allows us to learn $\hat{r}_\phi(x, \tau)$ via a binary classification problem: maximizing the likelihood of the observed preferences,

$$\max_\phi \quad \mathbb{E}_{(\tau_1, \tau_2, y) \sim \mathcal{D}_p} \left[ y \log p_\phi(\tau_1 \succ \tau_2) \right].$$

Once $\hat{r}_\phi$ is learned, a policy $\pi_\theta$ can be optimized under this learned reward, ensuring $\tau_1 \succ \tau_2 \implies \pi_\theta(\tau_1) > \pi_\theta(\tau_2)$, so that preferred trajectories receive higher probabilities under the policy.

A major challenge in PbRL is obtaining reliable preference data. Labels often depend on expert judgment, leading to potential *preference conflicts*, such as cyclic preferences $\tau_1 \succ \tau_2$, $\tau_2 \succ \tau_3$, and $\tau_3 \succ \tau_1$. These contradictions violate transitivity, making it critical to construct consistent preference labels to ensure stable policy learning.

### 3.3. Derivation of Preference Optimization

The key insight of our method is to transform the quantitative reward signals into qualitative preferences. This transformation stabilizes learning process by avoiding the dependency on numerical reward signals and consistently emphasizes optimality. We begin our derivation with the entropy-regularized RL introduced by (Haarnoja et al., 2017), which was originally designed to encourage exploration.

A challenge in applying RL to COPs is the exponential growth of the state and action spaces with problem size, making efficient exploration difficult. A common approach to encourage exploration is to include an entropy regularization term $\mathcal{H}(\pi_\theta)$ to balance exploitation and exploration:

$$\max_{\pi_\theta} \ \mathbb{E}_{x \sim \mathcal{D}} \left[ \mathbb{E}_{\tau \sim \pi_\theta(\cdot|x)} \left[ r(x, \tau) \right] + \alpha \mathcal{H}\left( \pi_\theta(\cdot \mid x) \right) \right], \quad (3)$$

where $\alpha > 0$ controls the strength of the entropy regularization, and $\mathcal{H}\left( \pi_\theta(\cdot \mid x) \right) = -\sum_\tau \pi_\theta(\tau \mid x) \log \pi_\theta(\tau \mid x)$ is the entropy of the policy for instance $x$. However, computing the entropy term $\mathcal{H}(\pi_\theta)$ is intractable in practice due to the exponential number of possible trajectories.

Following prior works (Ziebart et al., 2008; Haarnoja et al., 2017), it is straightforward to show that the optimal policy to the maximum entropy-based objective in Eq. 3 admits an analytical form:

$$\pi^*(\tau \mid x) = \frac{1}{Z(x)} \exp\left( \alpha^{-1} r(x, \tau) \right), \qquad (4)$$

where the partition function $Z(x) = \sum_\tau \exp\left( \alpha^{-1} r(x, \tau) \right)$ normalizes the policy over all possible trajectories $\tau$. The detailed derivation is included in the Appendix D.1. Although the solution space of COPs is finite and the reward function $r(x, \tau)$ is accessible, computing the partition function $Z(x)$ is still intractable due to the exponential number of possible trajectories. This intractability makes it impractical to utilize the analytical optimal policy directly in practice.

The specific formulation of Eq. 4 implies that the latent reward function $\hat{r}(x, \tau)$ can be reparameterized in relation to the corresponding policy $\pi(\tau \mid x)$, analogous to the approach adopted in (Rafailov et al., 2024) for a KL-regularized objective and in (Hejna & Sadigh, 2024) within the inverse RL framework. Eq. 4 can thereby be rearranged to express the reward function in terms of its corresponding optimal policy $\pi$ for the entropy-regularized objective:

$$\hat{r}(x, \tau) = \alpha \log \pi(\tau \mid x) + \alpha \log Z(x). \qquad (5)$$

From Eq. 5, the grounding reward function $r$ can be explicitly expressed by the optimal policy $\pi^*$ of Eq. 3. Then we can relate preferences between trajectories directly to the policy probabilities. Specifically, the preference between two trajectories $\tau_1$ and $\tau_2$ can be modeled by projecting the difference in their rewards into a paired preference distribution. Note that this analytic expression naturally avoids intractable term $Z(x)$, since $Z(x)$ is a constant w.r.t. the $\tau$ and cancels out when considering reward differences.

Using preference models, by substituting Eq. 5 into Eq. 2, the preference probability between two trajectories is:

$$p(\tau_1 \succ \tau_2 \mid x) = f\left(\alpha\left[\log \pi(\tau_1 \mid x) - \log \pi(\tau_2 \mid x)\right]\right), \quad (6)$$

By leveraging this relationship, we transform the quantitative reward into qualitative preferences in terms of policy $\pi$. Next, we could approximate $\pi$ with parameterized $\pi_\theta$.

**Proposition 3.1.** *Let $\hat{r}(x, \tau)$ be a reward function consistent with the Bradley-Terry, Thurstone, or Plackett-Luce models. For a given reward function $\hat{r}'(x, \tau)$, if $\hat{r}(x, \tau) - \hat{r}'(x, \tau) = h(x)$ for some function $h(x)$, it holds that both $\hat{r}(x, \tau)$ and $\hat{r}'(x, \tau)$ induce the same optimal policy in the context of an entropy-regularized reinforcement learning problem.*

Based on Proposition 3.1, we can conclude that shifting the reward function by any function of the instance $x$ does not affect the optimal policy. This ensures that canceling out $Z(x)$ in Eq. 6 still preserves the optimality of the policy $\pi_\theta$ learned, we defer the proof to Appendix D.2.

**Comparison Criteria.** We adopt the grounding reward function $r$ to generate conflict-free preference labels $y = \mathbb{1}(\cdot) : \mathbb{R} \to \{0, 1\}$. As the reward function $r(x, \tau)$ in COPs can be seen as a physical measure, pairwise comparisons generated in this manner preserve a consistent and transitive partial order of preferences throughout the dataset. Moreover, while traditional RL methods may rely on affine transformations to scale the reward signal, our approach benefits from the affine invariance of the preference labels. Specifically, the indicator function is invariant under positive affine transformations: $\mathbb{1}(k \cdot r(x, \tau_1) + b > k \cdot r(x, \tau_2) + b) = \mathbb{1}(r(x, \tau_1) > r(x, \tau_2))$, for any $k > 0$ and any real number $b$. This property implies that our method emphasizes optimality independently of the scale and shift of the explicit reward function (e.g., reward shaping), facilitating the learning process by focusing on the relative superiority among solutions rather than their absolute reward values.

**Objective.** To make the approach practical, we approximate the optimal policy $\pi^*$ with a parameterized policy $\pi_\theta$. This approximation allows us to reparameterize the latent reward differences using $\pi_\theta$, naturally transforming the policy optimization into a classification problem analogous to the reward function trained in PbRL. Guided by the preference information from the grounding reward function $r(x, \tau)$, the optimization objective $J(\theta)$ can be formulated as:

$$\max_\theta \mathop{\mathbb{E}}_{x \sim \mathcal{D}, \tau \sim \pi_\theta(\cdot \mid x)} \left[\mathbb{1}\left((r(x, \tau_1) > r(x, \tau_2)) \cdot \log p_\theta(\tau_1 \succ \tau_2 \mid x)\right], \quad (7)$$

while instantiating with BT model $\sigma(\cdot)$, maximizing $p(\tau_1 \succ \tau_2 \mid x) = \sigma(\hat{r}_\theta(x, \tau_1) - \hat{r}_\theta(x, \tau_2))$ leads to the gradient:

$$\nabla_\theta J(\theta) \approx \frac{\alpha}{|\mathcal{D}||S_x|^2} \sum_{x \in \mathcal{D}} \sum_{\tau \in S_x} \sum_{\tau' \in S_x} [(g_{\text{BT}}(\tau, \tau', x) - g_{\text{BT}}(\tau', \tau, x)) \nabla_\theta \log \pi_\theta(\tau \mid x)] \quad (8)$$

where $g_{\text{BT}}(\tau, \tau', x) = \mathbb{1}(r(x, \tau) > r(x, \tau')) \cdot \sigma(\hat{r}_\theta(x, \tau') - \hat{r}_\theta(x, \tau))$, and $\hat{r}_\theta(x, \tau)) = \alpha \log \pi_\theta(\tau \mid x) + \alpha \log Z(x)$. Taking a deeper look at the gradient level, compared to the REINFORCE-based algorithm in Eq. 1, the term about $g(\tau, \tau', x) - g(\tau', \tau, x)$ serves as a quantity-invariant advantage signal. A key finding is that this reparameterized reward signal ensures that if $r(x, \tau_1) > r(x, \tau_2)$, then the gradient will favor increasing $\pi_\theta(\tau_1)$ over $\pi_\theta(\tau_2)$.

---

**Algorithm 1** Preference Optimization for COPs.

---

**Input:** problem set $\mathcal{D}$, number of training steps $T$, fine-tune steps $T_{\text{FT}} \geq 0$, batch size $B$, learning rate $\eta$, ground truth reward function $r$, number of local search iteration $I_{\text{LS}}$, initialized policy $\pi_\theta$.

**for** $step = 1, \ldots, T + T_{\text{FT}}$ **do**

  *//Sampling $N$ solutions for each instance $x_i$*

  $x_i \leftarrow \mathcal{D}, \quad \forall i \in \{1, \ldots, B\}$

  $\tau_i = \{\tau_i^1, \tau_i^2, \ldots, \tau_i^N\} \leftarrow \pi_\theta(x_i), \forall i \in \{1, \ldots, B\}$

  *// Fine-tuning with LS for $T_{\text{FT}}$ steps* **(Optional)**

  **if** $step > T$ **then**

    $\{\hat{\tau}_i^1, \hat{\tau}_i^2, \ldots, \hat{\tau}_i^N\} \leftarrow LocalSearch(\tau_i, r, I_{\text{LS}}), \forall i$

    $\tau_i \leftarrow \tau_i \cup \{\hat{\tau}_i^1, \hat{\tau}_i^2, \ldots, \hat{\tau}_i^N\}$

  **end if**

  *//Calculate conflict-free preference labels via grounding reward function $r(x, \tau)$*

  $y_{j,k}^i \leftarrow \mathbb{1}\left(r(x_i, \tau_i^j) > r(x_i, \tau_i^k)\right), \forall j, k$

  *//Approximating the gradient according to Eq. 8*

  $\nabla_\theta J(\theta) \leftarrow \dfrac{\alpha}{B|\tau_i|^2} \sum\limits_{i=1}^{B} \sum\limits_{j=1, k=1}^{|\tau_i|} \Big[\big(g(\tau_i^j, \tau_i^k, x_i) -$

    $g(\tau_i^k, \tau_i^j, x_i)\big) \nabla_\theta \log \pi_\theta(\tau_i^j \mid x_i)\Big]$

  $\theta \leftarrow \theta + \eta \nabla_\theta J(\theta)$

**end for**

---

### 3.4. Fine-Tuning with Local Search

Although neural solvers offer computational efficiency, they often struggle to achieve near-optimal solutions compared to the heuristic solvers. After policy convergence, additional standard training fails to improve the policy's performance, which can be observed in both PO and REINFORCE-based algorithms within the RL4CO framework.

Local Search (LS) is commonly employed as a post-processing step to refine solutions generated by combinatorial solvers, guaranteeing monotonic improvement. Specifically, for any solution $\tau$, the refined solution $\text{LS}(\tau)$ satisfies $r(x, \text{LS}(\tau)) \geq r(x, \tau)$ through localized adjustments. When integrating it into learning, traditional RL algorithms typically depend on numerical reward magnitudes, which may exhibit small variations following LS and thus yield weak gradient signals. In contrast, PO framework relies on qualitative comparisons rather than numerical reward values,

Table 1: Experiment results on TSP and CVRP. Gap is evaluated on 10k instances and Times are summation of them.

| | Solver | Algorithm | TSP | | | | | | CVRP | | | | | |
|---|---|---|---|---|---|---|---|---|---|---|---|---|---|---|
| | | | $N = 20$ | | $N = 50$ | | $N = 100$ | | $N = 20$ | | $N = 50$ | | $N = 100$ | |
| | | | Gap | Time | Gap | Time | Gap | Time | Gap | Time | Gap | Time | Gap | Time |
| **Heuristic** | Concorde | - | 0.00% | 13m | 0.00% | 21.5m | 0.00% | 1.2h | - | - | - | - | - | - |
| | LKH3 | - | 0.00% | 28s | 0.00% | 4.3m | 0.00% | 15.6m | 0.09% | 0.5h | 0.18% | 2h | 0.55% | 4h |
| | HGS | - | - | - | - | - | - | - | 0.00% | 1h | 0.00% | 3h | 0.00% | 5h |
| **Neural Solvers** | AM (Kool et al., 2019) | RF | 0.28% | 0.1s | 1.66% | 1s | 3.40% | 2s | 4.40% | 0.1s | 6.02% | 1s | 7.69% | 3s |
| | | PO | 0.33% | 0.1s | 1.56% | 1s | 2.86% | 2s | 4.60% | 0.1s | 5.65% | 1s | 6.82% | 3s |
| | Pointerformer (Jin et al., 2023) | RF | **0.00%** | 6s | 0.02% | 12s | 0.15% | 1m | - | - | - | - | - | - |
| | | PO | **0.00%** | 6s | 0.01% | 12s | 0.06% | 1m | - | - | - | - | - | - |
| | Sym-NCO (Kim et al., 2022) | RF | 0.01% | 1s | 0.16% | 2s | 0.39% | 8s | 0.72% | 1s | 1.31% | 4s | 2.07% | 16s |
| | | PO | **0.00%** | 1s | 0.08% | 2s | 0.28% | 8s | 0.63% | 1s | 1.20% | 4s | 1.88% | 16s |
| | POMO (Kwon et al., 2020) | RF | 0.01% | 1s | 0.04% | 15s | 0.15% | 1m | 0.37% | 1s | 0.94% | 5s | 1.76% | 3.3m |
| | | PO | **0.00%** | 1s | 0.02% | 15s | 0.07% | 1m | 0.16% | 1s | 0.68% | 5s | 1.37% | 3.3m |
| | | PO+Finetune | **0.00%** | 1s | **0.00%** | 15s | **0.03%** | 1m | **0.08%** | 1s | **0.53%** | 5s | **1.19%** | 3.3m |

making it inherently compatible with LS-driven fine-tuning.

To harness the benefits of LS while avoiding additional inference costs, we integrate LS into the *post-training* process (i.e. fine-tuning) after the policy has converged through standard RL training procedures. For each generated solution $\tau$, we apply a small number of LS iterations to obtain an improved solution $\text{LS}(\tau)$. In practice, $r(x, \text{LS}(\tau)) > r(x, \tau)$ for most cases except in the case that LS fails to improve.

We then form a preference tuple $(\tau, \text{LS}(\tau), y)$, where $y = \mathbb{1}\big(r(x, \text{LS}(\tau)) > r(x, \tau)\big)$. The optimization objective during fine-tuning becomes:

$$
\begin{aligned}
\max_{\theta} \quad & \mathbb{E}_{x \sim \mathcal{D}, \, \tau \sim \pi_\theta(\cdot|x)} \Big[ y \cdot \log p_\theta\big(\text{LS}(\tau) \succ \tau \mid x\big) \Big] \\
= \quad & \mathbb{E}_{x \sim \mathcal{D}, \, \tau \sim \pi_\theta(\cdot|x)} f\left( \alpha \left[ \log \pi_\theta(\text{LS}(\tau)) \mid x \right] - \log \pi_\theta(\tau \mid x) \right),
\end{aligned}
\tag{9}
$$

where the LS-refined solutions act as high-quality preference references. By incorporating these references directly into training, the solver can internalize local search improvements rather than relying on LS as a separate, computationally expensive post-processing step. Although each LS iteration introduces some overhead, it remains manageable by limiting the number of LS calls per trajectory. As summarized in Algorithm 1, the resulting algorithm seamlessly combines neural policy learning and local search in a unified, end-to-end training pipeline.

A potential concern lies in the off-policy nature of LS-refined solutions, since existing REINFORCE-based algorithms require on-policy sampling. When adopting such a fine-tuning process into REINFORCE, distribution shifts introduced by LS could demand importance sampling to correct for. However, during fine-tuning, PO's preference-based objective naturally aligns with an imitation learning perspective (e.g., BC (Pomerleau, 1988) and DAgger (Ross

et al., 2011)), treating LS outputs as expert demonstrations. Thus, PO can absorb these LS-refined solutions without encountering severe off-policy issues, allowing the policy to imitate these expert-like trajectories effectively.

## 4. Experiments

In this section, we present the main results of our experiments, demonstrating the superior performance of the proposed Preference Optimization (PO) algorithm for COPs. We aim to answer the following questions: 1. *How does PO compare to existing algorithms on standard benchmarks such as the Traveling Salesman Problem (TSP), the Capacitated Vehicle Routing Problem (CVRP) and the Flexible Flow Shop Problem (FFSP)?* 2. *How effectively does PO balance exploitation and exploration by considering entropy, in comparison to traditional RL algorithms?*

**Benchmark Setups.** We implement the PO algorithm upon various RL-based neural solvers, emphasizing that it is a general algorithm not tied to a specific model structure. The fundamental COPs including TSP, CVRP and FFSP, serve as our testbed. In routing problems, the reward $r(x, \tau)$ is defined as the Euclidean length of the trajectory $\tau$. The TSP aims to find a Hamiltonian cycle on a graph, minimizing the trajectory length, while the CVRP incorporates capacity constraints for vehicles and points, along with a depot as the starting point. Our main experiments utilize problems sampled from a uniform distribution, as prescribed in (Kool et al., 2019). The experiments on the FFSP are conducted to schedule tasks across multiple stages of machines with the objective of minimizing the makespan (MS.), which refers to the total time required for completing all tasks. To evaluate generalization capabilities, we conduct zero-shot testing on problems with diverse distributions as specified

Table 2: Experiment results on FFSP. MS and Gap are evaluated on 1k instances, Time are summation of them. * indicate the results are sourced from MatNet (Kwon et al., 2021) and Aug. indicate the data augmentation for inference.

| | Solver | FFSP20 | | | FFSP50 | | | FFSP100 | | |
|---|---|---|---|---|---|---|---|---|---|---|
| | | MS. ↓ | Gap | Time | MS. ↓ | Gap | Time | MS. ↓ | Gap | Time |
| Heuristic | CPLEX (60s)* | 46.4 | 84.13% | 17h | | × | | | × | |
| | CPLEX (600s)* | 36.6 | 45.24% | 167h | | × | | | × | |
| | Random | 47.8 | 89.68% | 1m | 93.2 | 88.28% | 2m | 167.2 | 87.42% | 3m |
| | Shortest Job First | 31.3 | 24.21% | 40s | 57.0 | 15.15% | 1m | 99.3 | 11.33% | 2m |
| | Genetic Algorithm | 30.6 | 21.43% | 7h | 56.4 | 13.94% | 16h | 98.7 | 10.65% | 29h |
| | Particle Swarm Opt. | 29.1 | 15.48% | 13h | 55.1 | 11.31% | 26h | 97.3 | 9.09% | 48h |
| Neural Solver | MatNet (RF) | 27.3 | 8.33% | 8s | 51.5 | 4.04% | 14s | 91.5 | 2.58% | 27s |
| | MatNet (PO) | 27.0 | 7.14% | 8s | 51.3 | 3.64% | 14s | 91.1 | 2.13% | 27s |
| | MatNet (RF+Aug.) | 25.4 | 0.79% | 3m | 49.6 | 0.20% | 8m | 89.7 | 0.56% | 23m |
| | MatNet (PO+Aug.) | **25.2** | **0.00%** | 3m | **49.5** | **0.00%** | 8m | **89.2** | **0.00%** | 23m |

in (Bi et al., 2022) and on standard benchmark datasets: TSPLib (Reinelt, 1991) and CVRPLib (Uchoa et al., 2017). The hyperparameter configurations for these solvers primarily follow their original implementations, with detailed specifications provided in Appendix E.3. Most experiments were conducted on a server with NVIDIA 24GB-RTX 4090 GPUs and an Intel Xeon Gold 6133 CPU.

**Baselines.** We employ well-established heuristic solvers, including LKH3 (Helsgaun, 2017), HGS (Vidal, 2022), Concorde (Applegate et al., 2006) for routing problems and CPLEX (Cplex, 2009) for FFSP, to evaluate the optimality gap. We also compare against RL-based neural solvers that use variants of REINFORCE: AM (Kool et al., 2019), POMO (Kwon et al., 2020), Sym-NCO (Kim et al., 2022), Pointerformer (Jin et al., 2023) and ELG (Gao et al., 2024) for TSP/CVRP, and MatNet (Kwon et al., 2021) for FFSP. Additional experiments on large scale COPs with hybrid solver DIMES (Qiu et al., 2022) are included in Appendix F.1. Since all these neural solvers are originally trained using modified REINFORCE methods, we collectively refer to these algorithms as **RF(s)** for simplicity. AM estimates the advantage function using its previous step solver, while POMO refines this by averaging the quality of sampled solutions from the same instance as a baseline. Building upon POMO, Sym-NCO further improves REINFORCE by leveraging problem equivalences for data augmentation, and Pointerformer enhances stability through reward normalization (i.e., reward shaping).

**Choice of parameter $\alpha$.** The parameter $\alpha$ in PO framework stems from the entropy-regularized objective, which governs the exploration-exploitation trade-off during training. Higher $\alpha$ values increase entropy regularization, thereby promoting exploration of the solution space, while lower values emphasize exploitation. In our implementation, we systematically adopt lower $\alpha$ values for training the solvers that incorporate built-in exploration mechanisms (e.g., POMO, Sym-NCO, and Pointerformer), as these architectures already maintain sufficient diversity in their solution sampling. This calibration prevents excessive exploration that could impede convergence. Further details regarding our parameter tuning methodology are provided in Appendix E.2.

### 4.1. Comparison with Existing Algorithms on Standard Benchmarks

We aim to compare the proposed Preference Optimization (**PO**) method with existing modified REINFORCE (termed as **RF**) methods, considering sample efficiency during training, solution quality during inference and generalization ability on unseen instances with different distributions and sizes on TSPLib and CVRPLib.

**Sample Efficiency.** The training performance of PO and RF upon POMO, Sym-NCO and Pointerformer are illustrated in Figure 2. PO achieves a convergence speed 1.5x to 2.5x faster than RFs on such solvers. Notably for POMO and Sym-NCO, training with PO for 80 epochs yields comparable performance to that with RF for 200 epochs. Similar improvements are observed for Pointerformer. For FFSP using MatNet and large-scale TSP using DIMES, PO achieves comparable performance with only 60%–70% training epochs to that of RFs. This demonstrates the inherent exploration ability of PO, which stem from entropy-regularized objective in Eq. 3. Additional experiments on COMPASS (Chalumeau et al., 2023) and Poppy (Grinsztajn et al., 2023) are included in Appendix F.3.

**Solution Quality.** As shown in Table 1, while sharing the same inference times, models trained with PO mostly outperform those trained with the RFs in terms of solution quality. We also perform fine-tuning with Local Search (2-Opt (Croes, 1958) for TSP and swap* (Vidal, 2022) for CVRP) as mentioned in Section 3.4. After few steps for fine-tuning, POMO achieves an gap of only 0.03% on TSP-100 and 1.19% on CVRP-100, demonstrating that when approaching the optimal solution, PO can further enhance

Table 3: Zero-shot generalization experiments on TSPLib and CVRPLib-Set-X benchmarks. Results show averaged computational time over entire testing set and optimality gaps across different problem size ranges. The SL and RL indicate Supervised Learning and Reinforcement Learning paradigms respectively.

| Solver | Paradigm | TSPLib | | | | CVRPLib | | | |
|---|---|---|---|---|---|---|---|---|---|
| | | (0, 200] Gap | (200, 1002] Gap | Total Gap | Time | (0, 200] Gap | (200, 1000] Gap | Total Gap | Time |
| LKH3 | Heuristic | 0.00% | 0.00% | 0.00% | 24s | 0.36% | 1.18% | 1.00% | 16m |
| HGS | Heuristic | - | - | - | - | 0.01% | 0.13% | 0.11% | 16m |
| NeuroLKH (Xin et al., 2021b) | Heuristic+SL | 0.00% | 0.00% | 0.00% | 24s | 0.47% | 1.16% | 0.88% | 16m |
| POMO (Kwon et al., 2020) | Neural Solver+RL | 3.07% | 13.35% | 7.45% | 0.41s | 5.26% | 11.82% | 10.37% | 0.80s |
| Sym-NCO (Kim et al., 2022) | | 2.88% | 15.35% | 8.29% | 0.34s | 9.99% | 27.09% | 23.32% | 0.87s |
| Omni-POMO (Zhou et al., 2023) | | 1.74% | 7.47% | 4.16% | 0.34s | 5.04% | 6.95% | 6.52% | 0.75s |
| Pointerformer (Jin et al., 2023) | | 2.31% | 11.47% | 6.32% | 0.24s | - | - | - | - |
| LEHD (Luo et al., 2023) | Neural Solver+SL | 2.03% | 3.12% | 2.50% | 1.28s | 11.11% | 12.73% | 12.25% | 1.67s |
| BQ-NCO (Drakulic et al., 2023) | | 1.62% | **2.39%** | **2.22%** | 2.85s | 10.60% | 10.97% | 10.89% | 3.36s |
| DIFUSCO (Sun & Yang, 2023) | | 1.84% | 10.83% | 5.77% | 30.68s | - | - | - | - |
| T2TCO (Li et al., 2023) | | 1.87% | 9.72% | 5.30% | 30.82s | - | - | - | - |
| ELG (RF) (Gao et al., 2024) | Neural Solver+RL | 1.12% | 5.90% | 3.08% | 0.63s | 4.51% | 6.46% | 6.03% | 1.90s |
| ELG (PO) | | **1.04%** | 5.84% | 3.00% | 0.63s | **4.39%** | **6.37%** | **5.94%** | 1.90s |

the policy by using expert knowledge to fine-tune. Moreover, we extended our evaluation to the FFSP. As summarized in Table 2, solvers trained with PO consistently achieve optimal compared to their RF counterparts and heuristic solvers. These results confirm that PO not only improves training efficiency but also leads to higher-quality solutions.

**Generalization Ability.** As PO is a general algorithmic improvement that can be applied to RL-based solvers, it could inherit the architectural advantages of existing neural solvers and enhance them. To empirically validate this, we adopt it to the ELG (Gao et al., 2024), which is specifically designed for generalization by incorporating a local policy. The zero-shot experiments on TSPLib and CVRPLib-Set-X in Table 3 demonstrate PO improves results in all cases compared with their original REINFORCE-based version ELG (RF). Further results about cross-distribution generalization experiments are included in Appendix F.5.

### 4.2. How Effectively does PO Balance Exploitation and Exploration?

**Consistency of Policy.** A key superiority of the proposed PO algorithm is its ability to consistently emphasize better solutions, independent of the numerical values of the advantage function. Figure 3a compares the advantage assignment between PO and REINFORCE-based algorithms. PO marginally separates high-quality trajectories by assigning them positive advantage values while allocating negative values to low-quality ones. In contrast, RFs struggles to differentiate trajectory quality, with most advantage values centered around zero. This distinction showcases PO's capability to both highlight superior solutions and suppress

inferior ones. Additionally, Figure 3b presents the distribution of advantage scales, where RFs exhibits a narrow, peaked distribution around zero, indicating limited differentiation. Conversely, PO-based methods display broader distributions, covering a wider range of both positive and negative values. This indicates PO's enhanced ability to distinguish between high- and low-quality trajectories, further supporting its effectiveness in solvers' learning process.

Furthermore, Figure 3c evaluates the consistency of the policies. PO significantly improves the consistency compared to RFs, and fine-tuning a pretrained solver with local search within PO framework further enhances consistency.

**Diversity for Exploration.** One limitation of existing REINFORCE-based algorithms is its incompatibility with entropy regularization at the trajectory level. In contrast, the PO method is derived from an entropy-regularized objective, which inherently promotes exploration. We compare the sum of entropy at each step during the early stage of training between PO and RF. As shown in Figure 3d, the model trained using PO achieves significantly higher entropy, indicating a more diverse set of explored strategies. On the other hand, the RF update scheme results in lower entropy, potentially leading to less efficient exploration. In conclusion, PO effectively balances exploration and exploitation, enabling the model to explore the solution space more thoroughly.

**Study on Preference Models.** A crucial aspect of PO is the choice of the preference model, as discussed in Section 3.3. Different preference models may lead to varying implicit reward models, as outlined in Eq. 7 and 8. Assuming a differentiable paired preference model $f(\cdot)$, the generalized form of the latent reward assigned for each

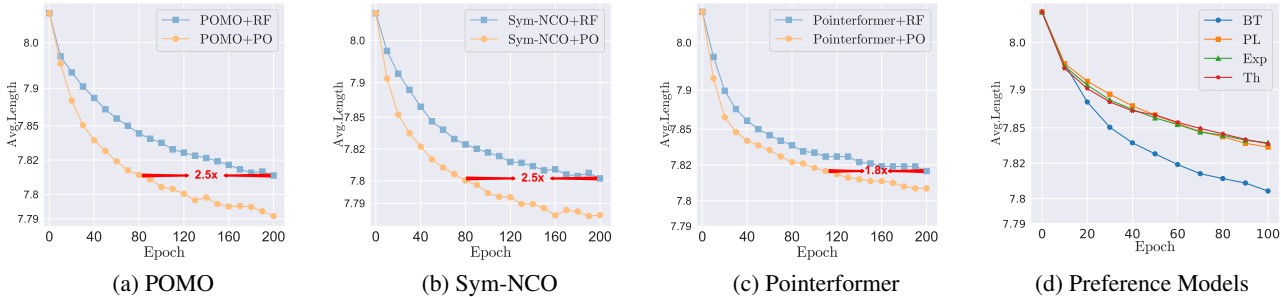

(a) POMO  (b) Sym-NCO  (c) Pointerformer  (d) Preference Models

Figure 2: **(a)-(c):** Comparison of PO and RFs on TSP-100 on different neural solvers; PO achieves RFs-level performance in only 40% - 60% training epochs, and surpasses RFs' solution quality consistently. **(d):** Comparison of different preference models: Bradley-Terry (BT), Plackett-Luce (PL), Thurstone (Th), and unbounded Exponential (Exp) (Azar et al., 2024).

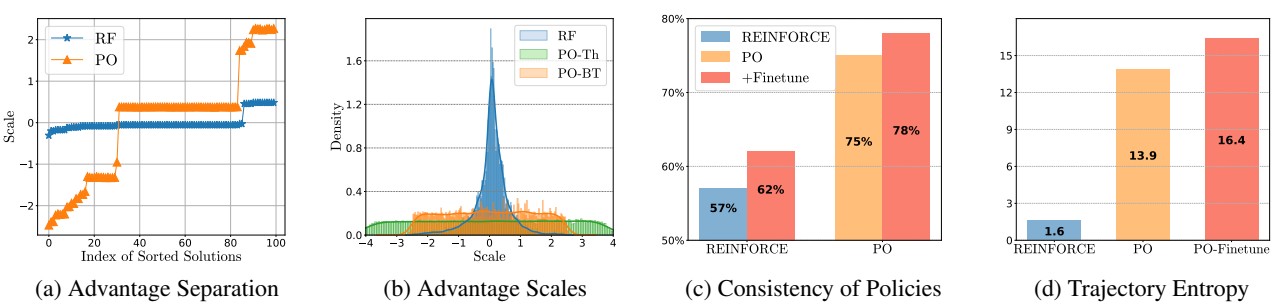

(a) Advantage Separation  (b) Advantage Scales  (c) Consistency of Policies  (d) Trajectory Entropy

Figure 3: **(a):** Advantage values for solutions sorted by their length, sampled from the trained model, PO significantly assigns separable advantage values than RF. **(b):** Distribution of advantage scales among different algorithms, comparing REINFORCE-based method, PO with the Thurstone model (PO-Th), and PO with the Bradley-Terry model (PO-BT). **(c):** Consistency measured as $p(\pi(\tau_1) > \pi(\tau_2) \mid r(\tau_1) > r(\tau_2))$. PO shows higher consistency than RF, with further improvement after fine-tuning. **(d):** Trajectory entropy, which is calculated as the sum of entropy at each step.

$\tau$ will be: $\frac{1}{|S_x|} \sum_{\tau' \in S_x} \left[ g_f(\tau, \tau', x) - g_f(\tau', \tau, x) \right]$, where $g_f(\tau, \tau', x) = \mathbb{1}\left(r(x, \tau) > r(x, \tau')\right) \cdot \frac{f'\left(\hat{r}_\theta(x, \tau) - \hat{r}_\theta(x, \tau')\right)}{f\left(\hat{r}_\theta(x, \tau) - \hat{r}_\theta(x, \tau')\right)}$ for any $\tau' \in S_x$. The results, shown in Figure 2d, indicate that the Bradley-Terry model outperforms the others on TSP-100. This suggests an interesting direction for further research, exploring the rationale behind the choice of preference models on different problems and their impact on the optimization landscape. More analyses and discussions are provided in Appendix F.4.

## 5. Conclusion

In this paper, we introduced **Preference Optimization**, a novel framework for solving COPs. By transforming quantitative reward signals into qualitative preference signals, PO addresses the challenges of diminishing reward differences and inefficient exploration inherent in traditional RL approaches. We naturally integrate PO with heuristic local search techniques into the fine-tuning process, enabling neural solvers to escape local optima during training and

generate near-optimal solutions without additional time during inference . Extensive experimental results demonstrate the practical viability and effectiveness of our approach, achieving superior sample efficiency and solution quality compared to common algorithms in RL4CO.

Notably, our work distinguishes itself from preference optimization methods in RLHF especially for LLMs in a critical dimension. While RLHF typically relies on *subjective*, offline human-annotated datasets, our Preference Optimization framework for COPs employs an active, online learning strategy grounded in *objective* metrics (e.g., route length) to identify and prioritize superior solutions.

Despite the promising results, we acknowledge several avenues for future research. The stability of our reparameterized reward function across diverse COPs warrants comprehensive investigation. Looking ahead, beyond COPs, applying PO to optimization problems where reward signals are difficult to design but preference information is readily available, such as multi-objective optimization, remains a valuable direction.

## Acknowledgment

We would like to express our sincere gratitude to the anonymous reviewers and (S)ACs of ICML 2025 for their thoughtful feedback and insightful comments, which have been instrumental in enhancing the quality of this work. We deeply appreciate their dedication to the scientific community. This work was supported by the National Key R&D Program of China (2022YFB4701400/4701402), SSTIC Grant(KJZD20230923115106012, KJZD202309231149160 32, GJHZ20240218113604008) and Beijing Key Lab of Networked Multimedia.

## Impact Statement

This paper presents work whose goal is to advance the field of Machine Learning. There may be many potential societal consequences of our work, none of which we feel must be specifically highlighted here.

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

# A. Combinatorial Optimization Problems: TSP and CVRP

We provide concise introductions to three fundamental combinatorial optimization problems: the Traveling Salesman Problem (TSP), the Capacitated Vehicle Routing Problem (CVRP) and the Flexible Flow Shop Problem (FFSP).

## A.1. Traveling Salesman Problem

The Traveling Salesman Problem (TSP) seeks to determine the shortest possible route that visits each city exactly once and returns to the origin city. Formally, given a set of cities $\mathcal{C} = \{c_1, c_2, \ldots, c_n\}$ and a distance matrix $D$ where $D_{i,j}$ represents the distance between cities $c_i$ and $c_j$, the objective is to find a trajectory $\tau = (c_1, c_2, \ldots, c_n, c_1)$ that minimizes the total travel distance:

$$\min_{\tau} \quad \sum_{k=1}^{n} D_{\tau(k), \tau(k+1)}.$$

Subject to:

$$\tau \text{ is a permutation of } \mathcal{C}, \quad \tau(n+1) = \tau(1).$$

Here, $\tau(k)$ denotes the $k$-th city in the trajectory, and the constraint $\tau(n+1) = \tau(1)$ ensures that the tour returns to the starting city.

## A.2. Capacitated Vehicle Routing Problem

The Capacitated Vehicle Routing Problem (CVRP) extends the TSP by introducing multiple vehicles with limited carrying capacities. The goal is to determine the optimal set of routes for a fleet of vehicles to deliver goods to a set of customers, minimizing the total distance traveled while respecting the capacity constraints of the vehicles.

Formally, given:

- A depot $c_0$,

- A set of customers $\mathcal{C} = \{c_1, c_2, \ldots, c_n\}$,

- A demand $d_i$ for each customer $c_i$,

- A distance matrix $D$ where $D_{i,j}$ represents the distance between locations $c_i$ and $c_j$,

- A fleet of $m$ identical vehicles, each with capacity $Q$,

the objective is to assign trajectories $\{\tau_1, \tau_2, \ldots, \tau_m\}$ to the vehicles such that each customer is visited exactly once, the total demand on any trajectory does not exceed the vehicle capacity $Q$, and the total distance traveled by all vehicles is minimized:

$$\min_{\{\tau_1, \tau_2, \ldots, \tau_m\}} \sum_{k=1}^{m} \sum_{l=1}^{|\tau_k|-1} D_{\tau_k(l), \tau_k(l+1)}.$$

Subject to:

$$\tau_k(1) = \tau_k(|\tau_k|) = c_0, \quad \forall k \in \{1, 2, \ldots, m\},$$
$$\bigcup_{k=1}^{m} \{\tau_k(2), \tau_k(3), \ldots, \tau_k(|\tau_k| - 1)\} = \mathcal{C},$$
$$\tau_k(i) \neq \tau_k(j) \quad \forall k \in \{1, 2, \ldots, m\}, \forall i \neq j,$$
$$\sum_{c_i \in \tau_k} d_i \leq Q, \quad \forall k \in \{1, 2, \ldots, m\}.$$

Here, $\tau_k(l)$ denotes the $l$-th location in the trajectory $\tau_k$ assigned to vehicle $k$. The constraints ensure that:

- Each trajectory starts and ends at the depot $c_0$.

- Every customer is visited exactly once across all trajectories.

- No customer is visited more than once within the same trajectory.

- The total demand served by each vehicle does not exceed its capacity $Q$.

### A.3. Trajectory Representation

In both TSP and CVRP, a trajectory $\tau$ represents a sequence of actions or decisions made by the policy to construct a solution. For TSP, $\tau$ is a single cyclic permutation of the cities, whereas for CVRP, $\tau$ comprises multiple routes, each assigned to a vehicle. Our Preference Optimization framework utilizes these trajectories to model and compare solution quality through preference signals derived from statistical comparison models.

### A.4. Flexible Flow Shop Problem

The Flexible Flow Shop Problem (FFSP) is a combinatorial optimization problem commonly encountered in scheduling tasks. It generalizes the classic flow shop problem by allowing multiple parallel machines at each stage, where jobs can be processed on any machine within a stage. The primary goal is to assign and sequence jobs across stages to minimize the makespan, which is the total time required to complete all jobs.

The optimization objective for FFSP can be mathematically formulated as:

$$\min_{\sigma, \mathbf{x}} C_{\max} = \max_{j \in \mathcal{J}} \left\{ C_j^{m_s} \right\},$$

subject to:

$$
\begin{aligned}
& C_j^{m_s} = S_j^{m_s} + p_j^{m_s}, \quad \forall j \in \mathcal{J}, \forall m_s \in \mathcal{M}, \\
& S_j^{m_s} \geq C_j^{m_{s-1}}, \quad \forall j \in \mathcal{J}, \forall m_{s-1} \in \mathcal{M}, \\
& S_j^{m_s} \geq C_{j'}^{m_s}, \quad \forall (j, j') \in \mathcal{J}, \text{ if } \sigma(j) > \sigma(j'), \\
& x_{j,m_s} = 1, \quad \text{if job } j \text{ is assigned to machine } m_s, \\
& \sum_{m_s \in \mathcal{M}} x_{j,m_s} = 1, \quad \forall j \in \mathcal{J}.
\end{aligned}
$$

Here: $\mathcal{J}$ is the set of jobs. $\mathcal{M}$ is the set of machines at each stage. $\sigma$ represents the sequence of jobs. $x$ is the assignment matrix of jobs to machines. $S_j^{m_s}$ is the start time of job $j$ on machine $m_s$. $C_j^{m_s}$ is the completion time of job $j$ on machine $m_s$. $p_j^{m_s}$ is the processing time of job $j$ on machine $m_s$. $C_{\max}$ is the makespan to be minimized.

The constraints ensure that jobs are scheduled sequentially on machines, maintain precedence, and adhere to the assignment rules. The FFSP is NP-hard and challenging to solve for large-scale instances.

## B. Graph Embedding and Solution Decoder

The Attention Model (AM)(Kool et al., 2019) represents a groundbreaking approach that successfully applied the Transformer architecture, based on attention mechanisms, to solve typical COPs such as routing and scheduling problems. AM adopts a classic Encoder-Decoder structure. Its core innovation lies in the Encoder, which utilizes a self-attention mechanism to comprehensively capture relationships between input nodes, while the Decoder employs a specialized attention mechanism (often a variant of Pointer Networks(Vinyals et al., 2015)) to sequentially construct solutions. Both POMO (Kwon et al., 2020) and Pointerformer (Jin et al., 2023) inherit this fundamental architecture.

### B.1. Graph Encoder

The encoder first projects raw features $\mathbf{x}_i \in \mathbb{R}^{d_x}$ (e.g., 2D coordinates) into initial node embeddings via a shared MLP:

$$\mathbf{h}_i^{(0)} = \text{MLP}(\mathbf{x}_i) \in \mathbb{R}^{d_h} \tag{10}$$

These initial embeddings are then refined through $N$ layers of multi-head attention (MHA). In each layer $l \in \{1, 2, \ldots, N\}$, the embedding for node $i$ is updated as follows:

$$\tilde{\mathbf{h}}_i^{(l)} = \mathbf{h}_i^{(l-1)} + \text{MHA}\left( \{\mathbf{h}_j^{(l-1)}\}_{j=1}^n \right) \tag{11}$$

$$\mathbf{h}_i^{(l)} = \tilde{\mathbf{h}}_i^{(l)} + \text{FFN}\left(\tilde{\mathbf{h}}_i^{(l)}\right) \tag{12}$$

where FFN represents a feed-forward network with activation.

To provide the Decoder with global contextual information about the entire problem instance, a graph-level embedding $\hat{\mathbf{h}}$ is computed by aggregating all final node embeddings. A common method is mean pooling:

$$\hat{\mathbf{h}} = \frac{1}{n} \sum_{i=1}^{n} \mathbf{h}_i^{(N)} \tag{13}$$

where $\mathbf{h}_i^{(N)}$ represents the final embedding of node $i$ after $N$ layers of attention, and $\hat{\mathbf{h}}$ captures the aggregate features of the entire graph.

### B.2. Solution Decoder

Following the encoding phase, a decoder is employed to sequentially generate the solution in an auto-regressive manner, modeling the COP as a Markov Decision Process. At each step $t$, it maintains a context vector $\mathbf{h}_t^{\text{ctx}}$ that represents the current state of the solution construction process. This context is initialized with the graph embedding and the first selected node (often a designated starting point, such as a depot in routing problems):

$$\mathbf{h}_0^{\text{ctx}} = [\hat{\mathbf{h}}; \mathbf{h}_{\tau_0}^{(N)}] \tag{14}$$

At subsequent steps $t > 0$, the context is updated to incorporate information about the most recently selected node:

$$\mathbf{h}_t^{\text{ctx}} = \text{MLP}(\mathbf{h}_0^{\text{ctx}}, \mathbf{h}_{\tau_{t-1}}^{(N)}) \tag{15}$$

The context-dependent query vector and node-specific key vectors are computed as:

$$\mathbf{q}_t = \mathbf{W}_Q^{\text{dec}} \mathbf{h}_t^{\text{ctx}} \tag{16}$$

$$\mathbf{k}_j = \mathbf{W}_K^{\text{dec}} \mathbf{h}_j^{(N)} \tag{17}$$

The compatibility between the current context and each potential next node is calculated using an attention-based scoring mechanism:

$$u_{tj} = \begin{cases} C \cdot \tanh\left(\frac{\mathbf{q}_t^\top \mathbf{k}_j}{\sqrt{d_h}}\right) & \text{if } j \in \mathcal{U}_t \\ -\infty & \text{otherwise,} \end{cases} \tag{18}$$

where $C$ is a temperature scaling parameter, $d_h$ is the dimension of the hidden representation, and $\mathcal{U}_t$ represents the set of feasible (typically unvisited) nodes at step $t$.

Finally, by applying the softmax function to the logits, the probability distribution for selecting the next node $\tau_t$ is obtained:

$$p(\tau_t = j \mid \mathbf{s}_t, \tau_{1:t-1}) = \frac{\exp(u_{tj})}{\sum_{k \in \mathcal{U}_t} \exp(u_{tk})} \tag{19}$$

### B.3. Model Variants and Extensions

Different architectural variants have been proposed to enhance the performance of attention-based neural solvers:

**POMO** (Kwon et al., 2020) leverages the inherent symmetry in many COPs by exploring multiple trajectories starting from different initial nodes. For a problem with nn n nodes, POMO generates nn n different solutions by starting the decoding process from each node, sharing parameters across all instances to improve training efficiency.

**Pointerformer** (Jin et al., 2023) enhances the encoder-decoder architecture with reversible residual network in Transformer blocks to effectively reduce the memory demand for larger-scale problems.

**Sym-NCO** (Kim et al., 2022) further exploits problem symmetry through sophisticated augmentation techniques, allowing the model to learn invariant representations that generalize better across different problem instances.

In all these variants, the core principle of encoding graph structure via attention mechanisms and decoding solutions via pointer-based selection remains consistent, demonstrating the flexibility and effectiveness of this paradigm for neural combinatorial optimization.

## C. Preference Models

In this section, we provide a concise overview of three widely used preference models: the Bradley-Terry (BT) model, the Thurstone model, and the Plackett-Luce (PL) model. These models are fundamental in statistical comparison modeling and form the basis for transforming quantitative reward signals into qualitative preference signals in our Preference Optimization (PO) framework.

### C.1. Bradley-Terry Model

The Bradley-Terry model is a probabilistic model used for pairwise comparisons. It assigns a positive parameter to each trajectory $\tau_i$, representing its preference strength. The probability that trajectory $\tau_i$ is preferred over trajectory $\tau_j$ is given by:

$$
\begin{aligned}
p(\tau_i \succ \tau_j) &= \frac{\exp(\hat{r}(\tau_i))}{\exp(\hat{r}(\tau_i)) + \exp(\hat{r}(\tau_j))} \\
&= \frac{1}{1 + \exp(-(\hat{r}(\tau_i) - \hat{r}(\tau_j)))} \\
&= \sigma(\hat{r}(\tau_i) - \hat{r}(\tau_j)).
\end{aligned}
$$

This model assumes that the preference between any two trajectories depends solely on their respective preference strengths, and it maintains the property of transitivity.

### C.2. Thurstone Model

The Thurstone model, also known as the Thurstone-Mosteller model, is based on the assumption that each trajectory $\tau_i$ has an associated latent score $s_i$, which is normally distributed. The probability that trajectory $\tau_i$ is preferred over trajectory $\tau_j$ is modeled as:

$$
p(\tau_i \succ \tau_j) = \Phi\left(\frac{\hat{r}(\tau_i) - \hat{r}(\tau_j)}{\sigma}\right),
$$

where $\Phi$ is the cumulative distribution function of the standard normal distribution, and $\sigma$ represents the standard deviation of the underlying noise. This model accounts for uncertainty in preferences and allows for probabilistic interpretation of comparisons. We adopt a normal distribution throughout this work.

### C.3. Plackett-Luce Model

The Plackett-Luce model extends pairwise comparisons to handle full rankings of multiple trajectories. It assigns a positive parameter $\lambda_i$ to each trajectory $\tau_i$, representing its utility. Given a set of trajectories to be ranked, the probability of observing a particular ranking $\tau = (\tau_1, \tau_2, \ldots, \tau_n)$ is given by:

$$
P(\tau) = \prod_{k=1}^{n} \frac{\exp(\hat{r}(\tau_k))}{\sum_{j=k}^{n} \exp(\hat{r}(\tau_j))}.
$$

This model is particularly useful for modeling complete rankings and can be extended to partial rankings. It preserves the property of independence of irrelevant alternatives and allows for flexible representation of preferences over multiple trajectories.

## D. Mathematical Derivations

### D.1. Deriving the Optimal Policy for Entropy-Regularized RL Objective

In this section, we derive the analytical solution for the optimal policy in an entropy-regularized reinforcement learning objective.

Starting from the entropy-regularized RL objective in Eq. 3:

$$
\max_{\pi} \quad \mathbb{E}_{x \sim \mathcal{D}} \left[ \mathbb{E}_{\tau \sim \pi_\theta(\cdot | x)} \left[ r(x, \tau) \right] + \alpha \mathcal{H}\left(\pi(\cdot \mid x)\right) \right],
$$

where $\mathcal{H}\left(\pi(\cdot \mid x)\right) = -\mathbb{E}_{\tau \sim \pi(\tau \mid x)}\left(\log \pi(\tau \mid x)\right)$ is the entropy of the policy, and $\alpha > 0$ is the regularization coefficient.

We can rewrite the objective as:

$$\max_{\pi} \quad \mathbb{E}_{x \sim \mathcal{D}, \ \tau \sim \pi(\tau \mid x)} \left[r(x, \tau) - \alpha \log \pi(\tau \mid x)\right]. \tag{20}$$

Our goal is to find the policy $\pi^*(\tau \mid x)$ that maximizes this objective. To facilitate the derivation, we can express the problem as a minimization:

$$\min_{\pi} \quad \mathbb{E}_{x \sim \mathcal{D}, \ \tau \sim \pi(\tau \mid x)} \left[\log \pi(\tau \mid x) - \frac{1}{\alpha} r(x, \tau)\right]. \tag{21}$$

Notice that:

$$\log \pi(\tau \mid x) - \frac{1}{\alpha} r(x, \tau) = \log \frac{\pi(\tau \mid x)}{\exp\left(\frac{1}{\alpha} r(x, \tau)\right)}. \tag{22}$$

Introduce the partition function $Z(x) = \sum_{\tau} \exp\left(\frac{1}{\alpha} r(x, \tau)\right)$, and define the probability distribution:

$$\pi^*(\tau \mid x) = \frac{1}{Z(x)} \exp\left(\frac{1}{\alpha} r(x, \tau)\right). \tag{23}$$

This defines a valid probability distribution over trajectories $\tau$ for each instance $x$, as $\pi^*(\tau \mid x) > 0$ and $\sum_{\tau} \pi^*(\tau \mid x) = 1$. Substituting Eq. 23 into Eq. 22, we have:

$$\log \pi(\tau \mid x) - \frac{1}{\alpha} r(x, \tau) = \log \frac{\pi(\tau \mid x)}{\pi^*(\tau \mid x)} + \log Z(x). \tag{24}$$

Therefore, the minimization problem in Eq. 21 becomes:

$$\min_{\pi} \mathbb{E}_{x \sim \mathcal{D}} \left[\mathbb{E}_{\tau \sim \pi(\tau \mid x)} \left[\log \frac{\pi(\tau \mid x)}{\pi^*(\tau \mid x)}\right] + \log Z(x)\right]. \tag{25}$$

Since $\log Z(x)$ does not depend on $\pi$, minimizing over $\pi$ reduces to minimizing the Kullback-Leibler (KL) divergence between $\pi(\tau \mid x)$ and $\pi^*(\tau \mid x)$:

$$\min_{\pi} \mathbb{E}_{x \sim \mathcal{D}} \left[D_{\mathrm{KL}}\left(\pi(\tau \mid x) \parallel \pi^*(\tau \mid x)\right)\right], \tag{26}$$

where the KL divergence is defined as:

$$\mathbb{D}_{\mathrm{KL}}\left(\pi(\tau \mid x) \parallel \pi^*(\tau \mid x)\right) = \mathbb{E}_{\tau \sim \pi(\tau \mid x)} \left[\log \frac{\pi(\tau \mid x)}{\pi^*(\tau \mid x)}\right].$$

The KL divergence is minimized when $\pi(\tau \mid x) = \pi^*(\tau \mid x)$ according to Gibbs' inequality. So, the optimal policy is:

$$\pi^*(\tau \mid x) = \frac{1}{Z(x)} \exp\left(\frac{1}{\alpha} r(x, \tau)\right). \tag{27}$$

This shows that the optimal policy under the entropy-regularized RL objective is proportional to the exponentiated reward function, normalized by the partition function $Z(x)$.

**Conclusion.** We have derived that the optimal policy $\pi^*(\tau \mid x)$ in the entropy-regularized RL framework is given by Eq. 27. This policy assigns higher probabilities to trajectories with higher rewards, balanced by the entropy regularization parameter $\alpha$, which controls the trade-off between exploitation and exploration.

### D.2. Proof of Proposition 3.1

**Proposition.** *Let $\hat{r}(x, \tau)$ be a reward function consistent with the Bradley-Terry, Thurstone, or Plackett-Luce models. For a given reward function $\hat{r}'(x, \tau)$, if there exists a function $h(x)$ such that $\hat{r}'(x, \tau) = \hat{r}(x, \tau) - h(x)$, then both $\hat{r}(x, \tau)$ and $\hat{r}'(x, \tau)$ induce the same optimal policy in the context of an entropy-regularized reinforcement learning problem.*

*Proof:* In an entropy-regularized reinforcement learning framework, the optimal policy $\pi^*(\tau \mid x)$ for a given reward function $\hat{r}(x, \tau)$ is given by:

$$\pi^*(\tau \mid x) = \frac{1}{Z(x)} \exp\left(\frac{1}{\alpha}\hat{r}(x, \tau)\right),$$

where $\alpha > 0$ is the temperature parameter (inverse of the regularization coefficient), and $Z(x)$ is the partition function defined as:

$$Z(x) = \sum_\tau \exp\left(\frac{1}{\alpha}\hat{r}(x, \tau)\right).$$

Similarly, for the reward function $\hat{r}'(x, \tau) = \hat{r}(x, \tau) - h(x)$, the optimal policy $\pi'^*(\tau \mid x)$ is:

$$\pi'^*(\tau \mid x) = \frac{1}{Z'(x)} \exp\left(\frac{1}{\alpha}\hat{r}'(x, \tau)\right) = \frac{1}{Z'(x)} \exp\left(\frac{1}{\alpha}[\hat{r}(x, \tau) - h(x)]\right), \tag{28}$$

where $Z'(x)$ is the partition function corresponding to $\hat{r}'(x, \tau)$:

$$Z'(x) = \sum_\tau \exp\left(\frac{1}{\alpha}\hat{r}'(x, \tau)\right) = \sum_\tau \exp\left(\frac{1}{\alpha}[\hat{r}(x, \tau) - h(x)]\right).$$

Simplifying the exponent in Eq. 28:

$$\exp\left(\frac{1}{\alpha}[\hat{r}(x, \tau) - h(x)]\right) = \exp\left(\frac{1}{\alpha}\hat{r}(x, \tau)\right)\exp\left(-\frac{1}{\alpha}h(x)\right).$$

Since $h(x)$ depends only on $x$ and not on $\tau$, the term $\exp\left(-\frac{1}{\alpha}h(x)\right)$ is a constant with respect to $\tau$. Therefore, we can rewrite Eq. 28 as:

$$\pi'^*(\tau \mid x) = \frac{1}{Z'(x)} \exp\left(-\frac{1}{\alpha}h(x)\right)\exp\left(\frac{1}{\alpha}\hat{r}(x, \tau)\right). \tag{29}$$

Combining constants:

$$\pi'^*(\tau \mid x) = \left(\frac{\exp\left(-\frac{1}{\alpha}h(x)\right)}{Z'(x)}\right)\exp\left(\frac{1}{\alpha}\hat{r}(x, \tau)\right).$$

Notice that the term $\frac{\exp\left(-\frac{1}{\alpha}h(x)\right)}{Z'(x)}$ is a normalization constant that ensures $\sum_\tau \pi'^*(\tau \mid x) = 1$. Similarly, for $\pi^*(\tau \mid x)$, the normalization constant is $\frac{1}{Z(x)}$.

Since both $\pi^*(\tau \mid x)$ and $\pi'^*(\tau \mid x)$ are proportional to $\exp\left(\frac{1}{\alpha}\hat{r}(x, \tau)\right)$, they differ only by their respective normalization constants. Therefore, they assign the same relative probabilities to trajectories $\tau$.

To formalize this, consider any two trajectories $\tau_1$ and $\tau_2$. The ratio of their probabilities under $\pi^*(\tau \mid x)$ is:

$$\frac{\pi^*(\tau_1 \mid x)}{\pi^*(\tau_2 \mid x)} = \frac{\exp\left(\frac{1}{\alpha}\hat{r}(x,\tau_1)\right)}{\exp\left(\frac{1}{\alpha}\hat{r}(x,\tau_2)\right)} = \exp\left(\frac{1}{\alpha}[\hat{r}(x,\tau_1) - \hat{r}(x,\tau_2)]\right). \tag{30}$$

Similarly, under $\pi'^*(\tau \mid x)$:

$$\frac{\pi'^*(\tau_1 \mid x)}{\pi'^*(\tau_2 \mid x)} = \frac{\exp\left(\frac{1}{\alpha}\hat{r}'(x,\tau_1)\right)}{\exp\left(\frac{1}{\alpha}\hat{r}'(x,\tau_2)\right)} = \exp\left(\frac{1}{\alpha}[\hat{r}'(x,\tau_1) - \hat{r}'(x,\tau_2)]\right). \tag{31}$$

Substituting $\hat{r}'(x,\tau) = \hat{r}(x,\tau) - h(x)$:

$$\hat{r}'(x,\tau_1) - \hat{r}'(x,\tau_2) = [\hat{r}(x,\tau_1) - h(x)] - [\hat{r}(x,\tau_2) - h(x)] = \hat{r}(x,\tau_1) - \hat{r}(x,\tau_2).$$

Therefore, the ratios in Eq. 30 and 31 are equal:

$$\frac{\pi^*(\tau_1 \mid x)}{\pi^*(\tau_2 \mid x)} = \frac{\pi'^*(\tau_1 \mid x)}{\pi'^*(\tau_2 \mid x)}.$$

Since the policies assign the same relative probabilities to all trajectories, and they are both properly normalized, it holds:

$$\pi^*(\tau \mid x) = \pi'^*(\tau \mid x), \quad \forall \tau.$$

Thus, $\hat{r}(x,\tau)$ and $\hat{r}'(x,\tau)$ induce the same optimal policy in the context of an entropy-regularized reinforcement learning problem. This result holds for the Bradley-Terry, Thurstone, and Plackett-Luce models because these models relate preferences to differences in reward values, and any constant shift $h(x)$ in the reward function does not affect the differences between reward values for different trajectories.

## E. Experiment Detail and Setting

### E.1. Implementation Details of the Code

The implementation of the Preference Optimization (PO) algorithm in Python using PyTorch is as follows:

```python
import torch.nn.functional as F

def preference_optimazation(reward, log_prob):
"""
    reward: reward for all solutions, shape(B, P)
    log_prob: policy log prob, shape(B, P)
"""
preference = reward[:, :, None] > reward[:, None, :]
log_prob_pair = log_prob[:, :, None] - log_prob[:, None, :]

# Under Brandley-Terry model:
pf_log = torch.log(F.sigmoid(self.alpha * log_prob_pair))

# Exponential: torch.log(torch.exp(self.alpha * log_prob_pair)):
pf_log = self.alpha * log_prob_pair

loss = -torch.mean(pf_log * preference)
return loss
```

### E.2. Parameter Tuning

**Methodological views.** From PO's entropy-regularized objective, the parameter $\alpha$ represents the exploration-exploitation trade-off. Higher $\alpha$ values promote exploration, while lower values emphasize exploitation. In our experiments, we employed a grid search within 0.005, 0.01, 0.05, 0.1, 0.5, 1.0, 2.0 for each problem-model combination.

**Empirical views.** Our empirical findings suggest that PO's performance is influenced by two additional factors:

*Network architecture's inherent exploration capacity:* Models with built-in exploration mechanisms (e.g., POMO and its variants with multi-start approaches) typically benefit from lower $\alpha$ values to prioritize exploitation but DIMES require more exploration with higher $\alpha$ values. For POMO and its variants on different problems, we observed that routing problems typically perform well with $\alpha$ in the range of 1e-2 to 1e-3, while FFSP benefits from $\alpha$ values between 1.0 and 2.0.

*Preference model selection:* As PO serves as a flexible framework, different preference models could yield distinct parameterized reward assignments, necessitating different $\alpha$ calibrations. The Exponential model could be a good candidate when the Bradley-Terry model underperforms on new problems, particularly for challenging problems, before exploring alternatives like Thurstone or Plackett-Luce models (which generalize Bradley-Terry beyond pairwise comparisons). Besides, we also provide detailed analyses regarding different preference models in Appendix F.4.

**Adapting to new problems.** For new applications, there are two intuitions for practical extensions:

*Length-control regularization:* For problems where sampled solutions have varying lengths and shorter solutions with lower costs are preferred, a length-control regularization factor $\frac{1}{|\tau|}$ can be effective, resulting in:

$$f(\alpha \left[ \frac{1}{|\tau_1|} \log \pi_\theta(\tau_1|x) - \frac{1}{|\tau_2|} \log \pi_\theta(\tau_2|x) \right]).$$

*Margin enhancement:* For models with limited capacity, a margin enhancement term $f(\alpha \left[ \log \pi_\theta(\tau_1|x) - \log \pi_\theta(\tau_2|x) \right] - \gamma)$ can help prioritize better solutions, where $\gamma$ serves as a margin parameter when $f(\cdot)$ is a non-linear function.

### E.3. Hyperparameters Setting

In our experimental setup, we set the tanh clip to 50.0 for VRPs, which has been shown to facilitate the training process (Jin et al., 2023). The following table presents the parameter settings for the four training frameworks: POMO (Kwon et al., 2020), Pointerformer (Jin et al., 2023), AM (Kool et al., 2019), and Sym-NCO (Kim et al., 2023).

**POMO** framework hyperparameter settings:

Table 4: Hyperparameter setting for POMO.

|  | TSP-100 | CVRP-100 |
| --- | --- | --- |
| Alpha | 0.05 | 0.03 |
| Preference Function | BT | Exponential |
| Epochs | 2000 | 4000 |
| Epochs (Finetune) | 100 | 200 |
| Epoch Size | 100000 | 50000 |
| Encoder Layer Number | 6 | 6 |
| Batch Size | 64 | 64 |
| Embedding Dimension | 128 | 128 |
| Attention Head Number | 8 | 8 |
| Feed Forward Dimension | 512 | 512 |
| Tanh Clip | 50 | 50 |
| Learning Rate | 3e-4 | 3e-4 |

Additional linear projection layer was adopted followed MHA in decoder as in (Kool et al., 2019).

**Pointerformer** framework hyperparameter settings:

Table 5: Hyperparameter setting for Pointerformer.

|                      | TSP   |
| -------------------- | ----- |
| Alpha                | 0.05  |
| Preference Function  | BT    |
| Epochs               | 2000  |
| Epoch Size           | 100000 |
| Batch Size           | 64    |
| Embedding Dimension  | 128   |
| Attention Head Number | 8    |
| Feed Forward Dimension | 512 |
| Encoder Layer Number | 6     |
| Learning Rate        | 1e-4  |

**AM** framework hyperparameter settings. Batch size of 512 contains 32 instances, each with 16 solutions, totaling 512 trajectories:

Table 6: Hyperparameter setting for AM.

|                      | TSP     | CVRP    |
| -------------------- | ------- | ------- |
| Alpha                | 0.05    | 0.03    |
| Preference Function  | BT      | BT      |
| Epochs               | 100     | 100     |
| Epoch Size           | 1280000 | 1280000 |
| Encoder Layer Number | 3       | 3       |
| Batch Size           | 256     | 256     |
| Embedding Dimension  | 128     | 128     |
| Attention Head Number | 8      | 8       |
| Tanh Clip            | 50      | 50      |
| Learning Rate        | 1e-4    | 1e-4    |

**Sym-NCO** framework hyperparameter settings:

Table 7: Hyperparameter setting for Sym-NCO.

|                      | TSP    | CVRP        |
| -------------------- | ------ | ----------- |
| Alpha                | 0.05   | 0.03        |
| Preference Function  | BT     | Exponential |
| Epochs               | 2000   | 4000        |
| Epoch Size           | 100000 | 50000       |
| Batch Size           | 64     | 64          |
| SR Size              | 2      | 2           |
| Embedding Dimension  | 128    | 128         |
| Attention Head Number | 8     | 8           |
| Feed Forward Dimension | 512  | 512         |
| Encoder Layer Number | 6      | 6           |
| Learning Rate        | 1e-4   | 1e-4        |

**DIMES** framework hyperparameter settings:

Table 8: Hyperparameter Setting for DIMES.

|  | TSP500 | TSP1000 | TSP10000 |
|---|---|---|---|
| Alpha | 2 | 2 | 2 |
| Preference Function | Exponential | Exponential | Exponential |
| KNN K | 50 | 50 | 50 |
| Outer Opt | AdamW | AdamW | AdamW |
| Outer Opt LR | 0.001 | 0.001 | 0.001 |
| Outer Opt WD | 1e-5 | 1e-5 | 1e-5 |
| Net Units | 32 | 32 | 32 |
| Net Act | SiLU | SiLU | SiLU |
| Emb Depth | 12 | 12 | 12 |
| Par Depth | 3 | 3 | 3 |
| Training Batch Size | 3 | 3 | 3 |

**MATNET** framework hyperparameter settings:

Table 9: Hyperparameter Setting for MATNET.

|  | FFSP20 | FFSP50 | FFSP100 |
|---|---|---|---|
| Alpha | 1.5 | 1.5 | 1 |
| Preference Function | Exponential | Exponential | Exponential |
| Pomo Size | 24 | 24 | 24 |
| Epochs | 100 | 150 | 200 |
| Epoch Size | 1000 | 1000 | 1000 |
| Encoder Layer Number | 3 | 3 | 3 |
| Batch Size | 50 | 50 | 50 |
| Embedding Dimension | 256 | 256 | 256 |
| Attention Head Number | 16 | 16 | 16 |
| Feed Forward Dimension | 512 | 512 | 512 |
| Tanh Clip | 10 | 10 | 10 |
| Learning Rate | 1e-4 | 1e-4 | 1e-4 |

## F. Additional Experiments.

### F.1. Experiments on Large Scale Problems

We further conduct experiments on large-scale TSP problems to validate the effectiveness of PO using the DIMES model (Qiu et al., 2022). DIMES leverages a reinforcement learning and meta-learning framework to train a parameterized heatmap, with REINFORCE as the optimization method in their original experiments. Solutions are generated by combining the heatmap with various heuristic methods, such as greedy decoding, MCTS, 2-Opt, or fine-tuning methods like Active Search (AS), which further train the solver for each instance.

As summarized in Table 10, our experiments demonstrate that PO improves the quality of the heatmap representations compared to REINFORCE. Across all decoding strategies (e.g., greedy, sampling, Active Search, MCTS (with 2-Opt as an inner loop)), PO-trained models consistently outperform their REINFORCE-trained counterparts in terms of solution quality, as evidenced by lower gap percentages across TSP500, TSP1000, and TSP10000. This confirms that PO enhances the learned policy, making it more effective regardless of the heuristic decoding method applied.

Table 10: Experiment results on large scale TSP.

| Method | TSP500 | | | TSP1000 | | | TSP10000 | | |
|---|---|---|---|---|---|---|---|---|---|
| | Len. ↓ | Gap | Time | Len. ↓ | Gap | Time | Len. ↓ | Gap | Time |
| LKH-3 | 16.55 | 0.00% | 46.28m | 23.12 | 0.00% | 2.57h | 71.79 | 0.00% | 8.8h |
| DIMES-G(RL) | 19.30 | 16.62% | 0.8m | 26.58 | 14.96% | 1.5m | 86.38 | 20.36% | 2.3m |
| DIMES-G(PO) | 18.82 | 13.73% | 0.8m | 26.22 | 13.39% | 1.5m | 85.33 | 18.87% | 2.3m |
| DIMES-S(RL) | 19.11 | 15.47% | 0.9m | 26.37 | 14.05% | 1.8m | 85.79 | 19.50% | 2.4m |
| DIMES-S(PO) | 18.75 | 13.29% | 0.9m | 26.07 | 12.74% | 1.8m | 85.21 | 18.67% | 2.4m |
| DIMES-AS(RL) | 17.82 | 7.68% | 2h | 24.99 | 8.09% | 4.3h | 80.68 | 12.39% | 2.5h |
| DIMES-AS(PO) | 17.78 | 7.42% | 2h | 24.73 | 6.97% | 4.3h | 80.14 | 11.64% | 2.5h |
| DIMES-MCTS(RL) | 16.93 | 2.30% | 3m | 24.30 | 5.10% | 6.3m | 74.69 | 4.04% | 27m |
| DIMES-MCTS(PO) | 16.89 | 2.05% | 3m | 24.33 | 5.23% | 6.3m | 74.61 | 3.93% | 27m |

## F.2. Training POMO for Long Time

Figure 4 compares the training efficiency of the PO and RL algorithms for TSP and CVRP. In the TSP task (a), PO reaches an objective value of 7.785 at epoch 400, while RL requires up to 1600 epochs to achieve comparable performance, demonstrating the sample efficiency of PO. This difference becomes more pronounced as training progresses. In the more challenging CVRP environment (b), PO continues to outperform RL, indicating its robustness and effectiveness in handling more complex optimization problems.

For TSP, each training epoch takes approximately 9 minutes, while each finetuning epoch with local search takes about 12 minutes. For CVRP, a training epoch takes about 8 minutes, and a finetuning epoch takes around 20 minutes. Since local search is executed on the CPU, it does not introduce additional GPU inference time. The finetuning phase constitutes 5% of the total epochs, adding a manageable overhead to the overall training time.

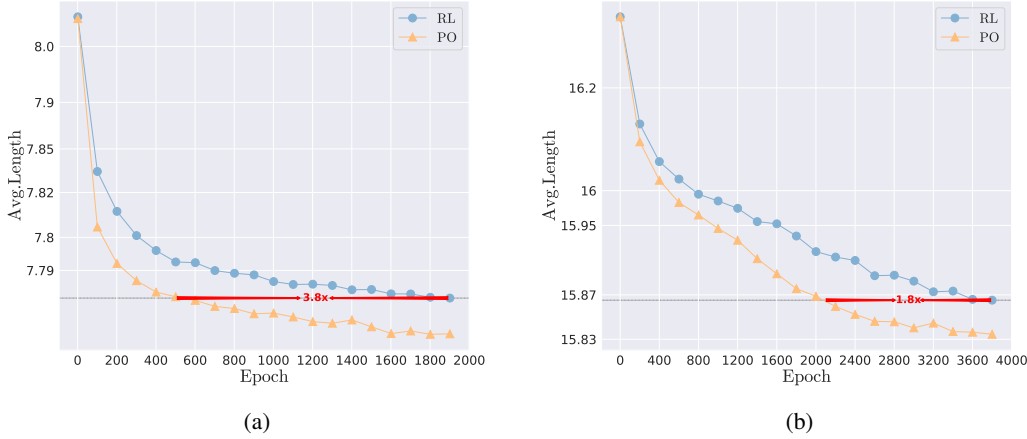

Figure 4: (a) Training curve for TSP (N=100) over 2000 epochs. (b) Training curve for CVRP (N=100) over 4000 epochs.

## F.3. Experiments on COMPASS and Poppy

To validate the PO's flexibility, we adapt it to the recent population-based framework COMPASS (Chalumeau et al., 2023) and Poppy (Grinsztajn et al., 2023). COMPASS learns a continuous latent space representing diverse strategies for combinatorial optimization and Poppy trains a population of reinforcement learning agents for combinatorial optimization, guiding unsupervised specialization via a "winner-takes-all" objective to produce complementary strategies that collaborate effectively to solve the problems. We implement PO upon these baselines and train them from scratch for 100k steps on a single 80GB-A800 GPU.

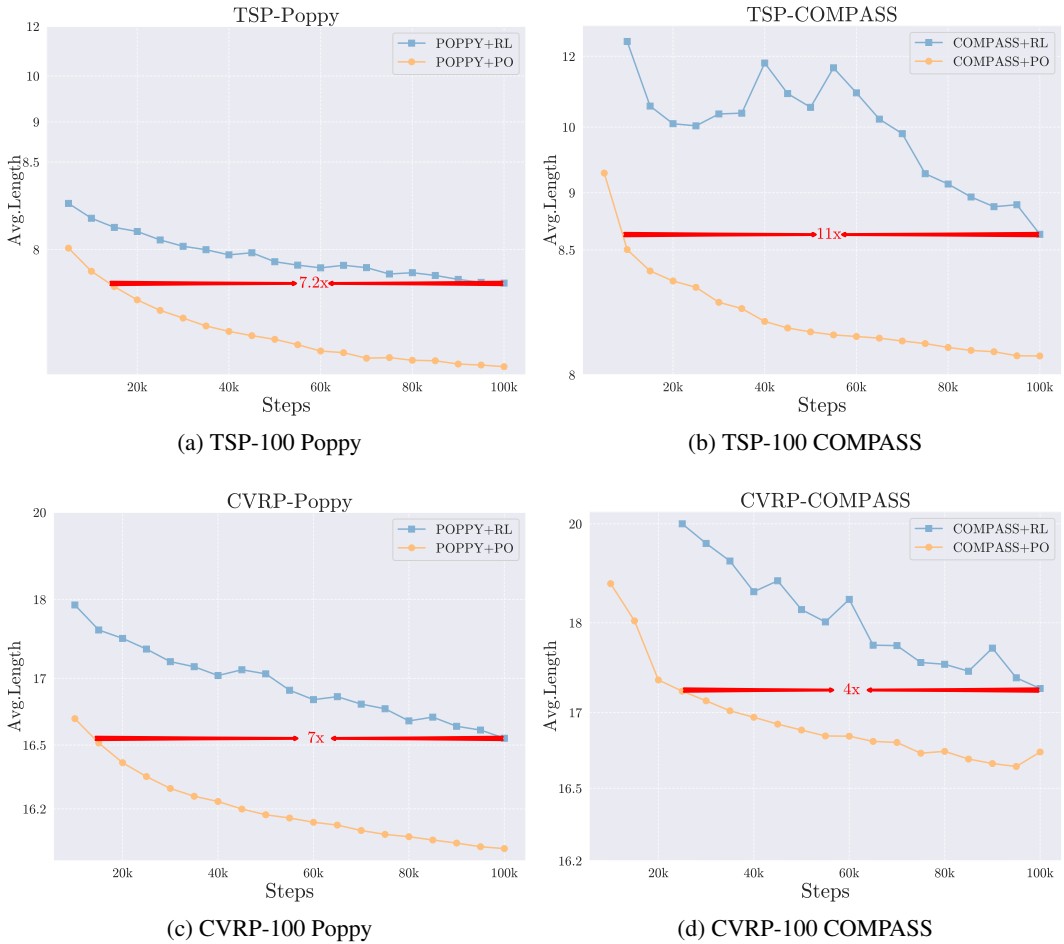

Figure 5: Training curves of PO and REINFORCE on Poppy and COMPASS.

The results above demonstrate that 1) PO significantly ensures lower optimality gap at the same iteration number; 2) PO ensures much faster convergence speed for the same gap, and higher stability during optimization. Moreover, these results also validate that our proposed algorithmic improvement method is consistently effective in various RL-based baselines.

### F.4. Preference Modeling

As indicated in (Azar et al., 2024), the Bradley-Terry and Thurstone models struggle to handle extreme scenarios where $p(\tau_1 \succ \tau_2) \approx 1$. Achieving such a near-certain preference requires $\hat{r}_\theta(x, \tau_1) - \hat{r}_\theta(x, \tau_2) \to +\infty$, implying $\frac{\pi_\theta(\tau_1)}{\pi_\theta(\tau_2)} \approx 0$. However, both the logistic function (in the Bradley-Terry model) and the CDF of the normal distribution (in the Thurstone model) exhibit gradient vanishing in this regime, as shown in Figure 6b. A natural alternative is the unbounded exponential function $f(\cdot) = \exp(\cdot)$, which places a much stronger emphasis on preferred solutions compared to Bradley-Terry and Thurstone.

Empirically, we observe that while the Bradley-Terry model performs better on smaller-scale problems (where overfitting is more likely, and a conservative preference function can mitigate this), the exponential function is more effective for larger-scale or complex (harder) COPs, as it prevents convergence to suboptimal local optima more vigorously. Future research could investigate theoretical properties of these preference models in different reward regimes, develop adaptive mechanisms that switch between them based on problem complexity.

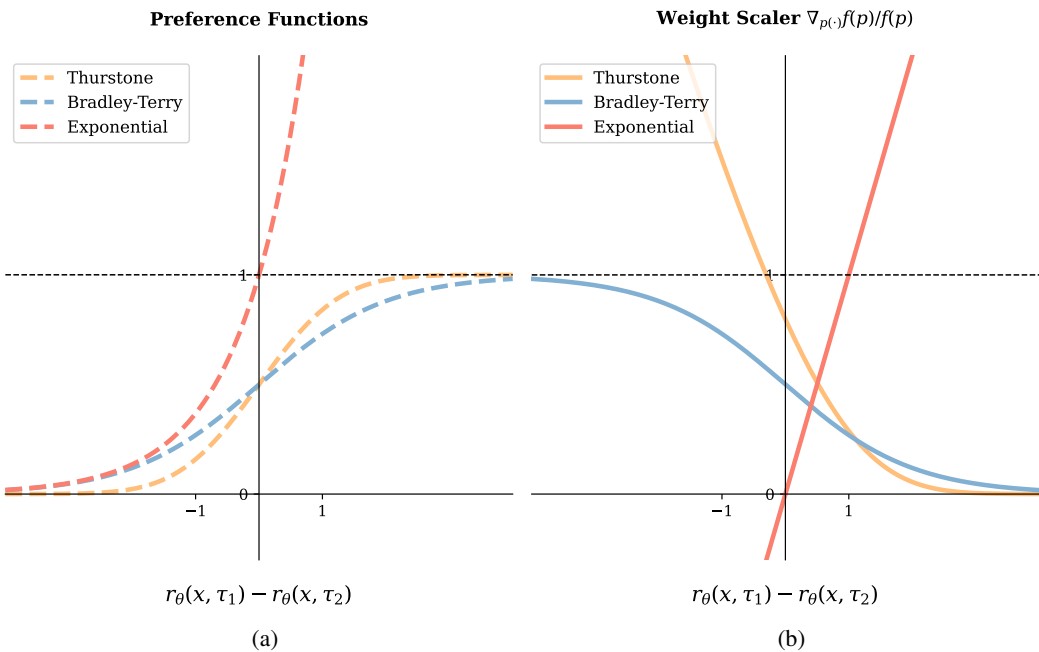

Figure 6: Comparison of three preference models—Thurstone, Bradley-Terry, and Exponential— under different reward differences $r_\theta(x, \tau_1) - r_\theta(x, \tau_2)$. (a) Illustrates how each model's preference function behaves as the reward gap changes. (b) Shows the corresponding weight (gradient) scaling factor, highlighting the gradient vanishing issue in Bradley-Terry and Thurstone for large positive reward gaps, and the strong emphasis provided by the exponential model.

## F.5. Generalization

We conducted a zero-shot cross-distribution evaluation, where models were tested on data from unseen distributions. Since models trained purely with RL tend to overfit to the training data distribution (Zhou et al., 2023), they may struggle with different reward functions in new distributions. However, training with PO helps mitigate this overfitting by avoiding the need for ground-truth reward signals. Following the diverse distribution setup in (Bi et al., 2022), the results are summarized in Table 11. Our findings show that the model trained with PO outperforms the original RL-based model across all scenarios.

Table 11: Zero-shot generalization experiment results. The Len and Gap are average on 10k instances.

|  | Method | Cluster | | Expansion | | Explosion | | Grid | | Implosion | |
|---|---|---|---|---|---|---|---|---|---|---|---|
|  |  | Len.↓ | Gap | Len.↓ | Gap | Len.↓ | Gap | Len.↓ | Gap | Len.↓ | Gap |
| TSP | LKH | 3.66 | 0.00% | 5.38 | 0.00% | 5.83 | 0.00% | 7.79 | 0.00% | 7.61 | 0.00% |
|  | POMO-RL | 3.74 | 2.09% | 5.41 | 0.60% | 5.85 | 0.20% | 7.80 | 0.16% | 7.63 | 0.15% |
|  | POMO-PO | **3.70** | **1.12%** | **5.40** | **0.34%** | **5.84** | **0.06%** | **7.79** | **0.04%** | **7.62** | **0.05%** |
| CVRP | HGS | 7.79 | 0.00% | 11.38 | 0.00% | 12.35 | 0.00% | 15.59 | 0.00% | 15.47 | 0.00% |
|  | POMO-RL | 7.97 | 2.28% | 11.51 | 1.29% | 12.48 | 0.97% | 15.79 | 0.86% | 15.60 | 0.87% |
|  | POMO-PO | **7.93** | **1.73%** | **11.49** | **1.12%** | **12.45** | **0.76%** | **15.76** | **0.63%** | **15.57** | **0.65%** |

