# OpenReview forum: "Preference Optimization for Combinatorial Optimization Problems"
_ICML.cc/2025/Conference — ICML 2025 poster_

### Official Review · Reviewer_HsDN · 2025-02-25

**Overall Recommendation:** 3

**Summary:**

The paper studies the problem of solving combinatorial optimization problems with reinforcement learning, focusing on two key issues: diminishing reward signals that slow down learning and inefficient exploration in large solution spaces. To address this, it introduces preference optimization, a method that replaces numerical rewards with preference-based signals. It also integrates local search into training rather than post-processing, aiming to improve solution quality without extra inference cost. The approach is tested on problems like the traveling salesman etc, claiming faster convergence and better solutions compared to standard RL methods.

**Claims And Evidence:**

In contribution 1, the authors mentioned optimal solutions. However, the paper is entirely empirical, I believe there is no evidence showing optimality.

**Essential References Not Discussed:**

I am not aware of any missing references.

**Experimental Designs Or Analyses:**

The experiment section is well presented without significant issue.

**Methods And Evaluation Criteria:**

Yes.

**Other Comments Or Suggestions:**

None

**Other Strengths And Weaknesses:**

Strengths:
- The paper presents experimental results on several combinatorial optimization problems, showing improvements over the baselines

Weaknesses:
- A key concern with this paper is the lack of a natural justification for applying preference learning to combinatorial optimization problems. In language models, preference learning arises organically because human feedback provides qualitative judgments on responses, making it a natural fit for reinforcement learning from human feedback. However, in COPs, there is no intrinsic notion of human preference—solutions are typically evaluated using well-defined numerical objective functions. It is unclear whether PO’s improvements stem from methodological advantages or better-tuned training procedures.

**Questions For Authors:**

1. Can the authors clarify what "fine-tuning" refers to in detail?
2. Can the authors provide a side-by-side comparison between the preference model used in this combinatorial optimization framework and other preference learning applications? I am particularly interested in understanding why preference models are the right tool for this setting. Unlike RLHF, where preference conflicts are often allowed, combinatorial optimization problems typically have well-defined objective functions. Given this distinction, how does preference learning in this work align with or differ from existing applications, and what makes it particularly suitable here?

**Relation To Broader Scientific Literature:**

The paper adapts preference-based learning, which is currently very popular in llm era, to combinatorial optimization problems. This can potentially open new chances for preference learning outside llm.

**Theoretical Claims:**

The paper does not include theoretical claims or proofs, except for a light proposition.

---

> ### Author Rebuttal · Authors · 2025-04-01
>
> We sincerely thank you for your thoughtful comments, which help us improve our presentation clarity. Below, we address your concerns regarding terminology and methodology.
>
> ---
> ### **1. Regarding the Claims of Optimality**
> We apologize for the confusion in Contribution 1 regarding *optimal solutions*. This term was used to indicate that our model preserves preference relations among candidate solutions—when solution $\tau_1$ is better than $\tau_2$ (the true relation), our model consistently assigns higher (implicit) reward value to $\tau_1$. This property is approximately ensured by preference modeling and shift-invariance. We will revise this description as *consistently emphasizing better solutions with relation preservation property* and justify this term in method section in detail.
>
> ---
> ### **2. Rationale of PO**
> + **Background: Reinforcement Learning for Combinatorial Optimization (RL4CO)**
> In the RL4CO paradigm, a neural policy iteratively samples a batch of feasible solutions and updates itself to favor better solutions according to the objective function (e.g., route length, makespan), naturally establishing the preference relationships. Our work focuses on *algorithmic improvements for training RL-based policy to enhance solution quality and learning efficiency* in RL4CO, which further improve the current REINFORCE-based algorithms that rely on numerical reward; this is one of the reasons for considering PO for COPs.
> + **Key Challenge: Diminishing Rewards**
> A critical issue in RL4CO is the diminishing reward signal. As solutions approach near-optimal, the numerical differences between their rewards become extremely small (e.g., 0.001), making it difficult for the policy trained with existing RL algorithms to distinguish better solutions from worse ones. The weakened signals cause premature convergence, preventing the policy from finding better solutions. Hopefully, PO can alleviate such issues due to the stability of preference relation, which is also the reason for choosing PO.
> + **Introducing Preference Optimization**
> To address this, we *convert the batch of numeric rewards into pairwise preferences*. Even if the reward difference is small, a preference-based approach gives a *categorical* label: B > A. This binary label can offer a more stable learning signal than the numerical one. Further, the learning process can be briefly summarized as:
> > 1. Sample a batch of feasible solutions from the current policy.
> > 2. Compare them by the objective function (e.g., cost, makespan) and label the better one as "preferred."
> > 3. Update the policy by *increasing the probability* of generating solutions that are "preferred (winner)" over others (loser) and back to Step 1.
>
> More details are provided in Algorithm 1 in our manuscript.
> + **Distinction from RLHF**
> 1. *Objective vs. Subjective Preferences*: In RLHF, human annotators may disagree or change their minds, creating conflicts. In PO, the grounding metric (e.g., route length) is naturally fixed, ensuring consistent and transitive preference signals.
> 2. *Online vs. Offline*: RLHF often uses a fixed dataset of human comparisons. In PO, our policy *actively* samples solutions, which are then compared by the objective function to produce preference labels
> 3. *Entropy vs. KL Regularization*: Our approach stems from an *entropy*-regularized RL objective, encouraging exploration in large discrete spaces. whereas RLHF typically employs KL-divergence to maintain proximity to a pretrained model.
>
> ---
> ### **3. Explanation of "Fine-Tuning"**
> In deep learning, *fine-tuning* typically means continuing to train an already-trained model with specialized data or new techniques to improve or adapt its performance. In our model, we similarly adopt this term to represent a two-stage training framework. Moreover, note that the stage 2 (fine-tune) is optional, and model with stage 1 already admit significant improvements over SOTA baselines; thus, the performance gain is indeed ensured by PO.
>
> In our context: 1. We *first train* a policy $\pi_\theta$ using PO. 2. Then, we *"fine-tune"* that policy by *incorporating local search into the training loop*.
>
> **Local search (LS)** is a standard technique in CO that makes small changes to a solution with non-degradation guarantee, which is generally fair to introduce this process into solver. The detailed fine-tune process can be summarized as:
> > 1. Take the policy's output $\tau$.
> > 2. Run local search on $\tau$ to get a slightly improved solution $\mathrm{LS}(\tau)$.
> > 3. Update the policy to favor $\mathrm{LS}(\tau)$ and go back to Step 1.
>
> This way, the policy *learns* from the local search refinements. This is what we refer to as "fine-tuning": continuing to refining the learned policy under the guidance of LS–based preference pairs.
>
> ---
> We appreciate the opportunity to clarify our work and will revise our manuscript accordingly. Thank you again for your valuable time and feedback and openly welcome any further discussions.

---

### Official Review · Reviewer_u9d3 · 2025-03-13

**Overall Recommendation:** 4

**Summary:**

This paper proposes a way to modify reinforcement learning (RL) so that it can deal with diminishing reward signals. The key idea is to turn the reward signals into pairwise preferences from which an underlying reward function can be learned. Another contribution of the paper is to use local search to generate additional pairwise preference relations (solution after local search is preferred over solution before local search).

The ideas are applied to combinatorial optimization problems, and it is shown that the conversion of reward to preference function is helpful on different problems (TSP, CVRP, FFSP) and for different underlying Neural RL approaches (AM, Pointerformer, Sym-NCO and POMO).

## update after rebuttal
The authors have addressed my questions and comments. I retain my score and recommend accepting the paper.

**Claims And Evidence:**

The main claims are
- that the proposed transformation of reward signals into preference signals addresses the issue of diminishing reward signals and inefficient exploration.
- That the reparameterized entropy-regularized objective bypasses the enumeration of the entire action space
- That the integration with local search helps during training.

The paper shows on different neural solvers that the transformation of reward signal into preference signal is beneficial. It is also demonstrated that the fine-tuning helps. However, here it is not clear whether the fine-tuning is done in addition or instead of the final iterations of the algorithm without fine-tuning. In the former case, one may wonder whether simply running the algorithm without fine-tuning for some additional generations would have yielded similar results.

**Essential References Not Discussed:**

I am not aware of missing references.

**Experimental Designs Or Analyses:**

It is not said explicitly, but I assume the training and test set are different (albeit drawn from the same distribution). Results seem to be based on single training runs, no standard errors are provided, which is not best practice.
However, given the consistency of the results across problems, it seems unlikely that replicating the experiments would lead to very different results. Thus, it does not raise a major concern in my eyes.

**Methods And Evaluation Criteria:**

The method is compared with the default method from the literature and with state-of-the-art heuristics such as LKH3 and HGS. The metrics used are optimality gap and inference time that are commonly used in the field.

**Other Comments Or Suggestions:**

NA

**Other Strengths And Weaknesses:**

- I understand the issue of diminishing reward signals. However, there are many possible solutions, e.g., simply replacing the reward by the rank of the solution. It is not intuitive why the proposed transformation into pairwise preferences is a natural choice. However, it seems to work.

- The name “Preference optimization” seems misleading, as it usually refers to optimization based on preference information where only such information is available, whereas in the current paper, the actual reward function is available but turned into preferences.

- It is not clear how parameters were chosen. I assume parameters for the baseline algorithms are taken from the corresponding original references? How were the alpha and the preference function chosen? How would they be chosen for a new problem (since different settings are chosen for different problems)?

- The paper doesn’t provide any information on e.g. the graph embedding or decoder used. It would be nice to have a summary in the appendix.

- While I believe it is not explicitly stated, the models are trained separately for different problem sizes. In practice, a heuristic that only works on a specific size seems very limited ted. Some experiments to see how the trained models generalize to other problem sizes would have been nice.

**Questions For Authors:**

1. Could you please confirm that the test set is different from the training set?

2. Could you clarify whether the fine-tuning is in addition to the non-fine-tuning iterations, or
replacing the last few non-fine-tuning iterations?

3. Do you have an idea how the approach generalizes across different sized problems?

**Relation To Broader Scientific Literature:**

The proposed mechanism is shown to improve results when integrated into three different methods from the literature, and tested on three different combinatorial problems. There is some relation to Preference-based RL which is discussed.

**Theoretical Claims:**

I did not review the mathematical derivations in Appendix C.

---

> ### Author Rebuttal · Authors · 2025-04-01
>
> We sincerely thank you for your affirmation on our work, and are grateful for the insightful review and constructive comments. We provide point-to-point responses as follows.
>
> ---
> ### **1. Math implications behind the proposed PO algorithm**
> We appreciate this insightful question on the rationale PO model, which can be justified from following aspects.
>
> *Transform numerical signals into ranks*. First, consider the REINFORCE algorithm:
> $$
> \nabla_{\theta} J(\theta)= \mathbb{E}\_{x \sim \mathcal{D}, \tau \sim \pi{\theta}(\tau \mid x)} \left[\left(r(x, \tau)-b(x)\right) \nabla_{\theta} \log \pi_{\theta}(\tau \mid x)\right].
> $$
> The policy is trained to increase the probability of solutions better than the baseline (where $r(x, \tau)-b(x) > 0$) and decrease the probability of worse solutions (where $r(x, \tau)-b(x) < 0$). This implies that *the policy mathematically acts as a ranking model that ranks solutions by their rewards*. Besides, explicitly learning the reward with traditional RL algorithms has two key weaknesses:
> 1) Sensitive to baseline selection;
> 2) Sensitive to the numerical scale of reward signals.
>
> *Transform ranks into preference relations*. Based on the math implication, we naturally connect the essential goal (discriminative ranking) with rank model. Since there are many possible solutions in rank, we adopt the preference model to retrieve the rank. Specifically, as long as the training pairs are sufficient, it is clear that the true ranks can be provably recovered by the pair-wise relations. Moreover, preference relations admit appealing properties, i.e., invariant to both baseline choice and the numerical scale of rewards. Thus, the PO-based model for ranks and numerical signals is generally reasonable from the views of methodology and numerical approximation.
>
> *The name 'PO'*. Note that PO here only means the methodology; thus, a more accurate description could be 'PO-based modeling for COPs'. We would clarify this point in revision.
>
> ---
> ### **2. Parameter setting**
> All parameters in baselines were adopted from their official codes, and we only retrained all models from scratch for comparison.
>
> ---
> ### **3. Choice of $\alpha$ in PO**
> We selected $\alpha$ through grid search with several training steps and found that the suitable value is both problem-dependent and model-dependent. This is why different $\alpha$ values are adopted for different problems and models.
>
> ---
> ### **4. Graph embedding and decoder**
> We utilize the same graph embedding and decoder as in the original works. We will include a comprehensive summary in the revision.
>
> ---
> ### **5. Generalization of PO**
> Methodologically, as PO is a general algorithmic improvement that can be applied to RL-based solvers, it could inherit the advantages of baselines and enhance them. To empirically validate this, we adopt it to the baselines, i.e., ELG[r1], that are specifically designed for generalization. The results on TSPLib and CVRPLib demonstrate this capability:
>
> **TSPLib:**
> |Method|(0, 200]|(200, 1002]|Total|Time|
> |-|-|-|-|-|
> |LKH3|0.00%|0.00%|0.00%|24s|
> |POMO|3.07%|13.35%|7.45%|0.41s|
> |Sym-POMO|2.88%|15.35%|8.29%|0.34s|
> |Omni-POMO|1.74%|7.47%|4.16%|0.34s|
> |Pointerformer|2.31%|11.47%|6.32%|0.24s|
> |LEHD|2.03%|3.12%|2.50%|1.28s|
> |BQ-NCO|1.62%|**2.39%**|**2.22%**|2.85s|
> |DIFUSCO|1.84%|10.83%|5.77%|30.68s|
> |T2TCO|1.87%|9.72%|5.30%|30.82s|
> |ELG-POMO (RF)|1.12%|5.90%|3.08%|0.63s|
> |ELG-POMO(PO)|**1.04%**|5.84%|3.00%|0.63s|
>
> **CVRPLib-Set-X:**
> |Method|(0, 200]|(200, 1000]|Total|Time|
> |-|-|-|-|-|
> |LKH3|0.36%|1.18%|1.00%|16m|
> |HGS|0.01%|0.13%|0.11%|16m|
> |POMO|5.26%|11.82%|10.37%|0.80s|
> |Sym-POMO|9.99%|27.09%|23.32%|0.87s|
> |Omni-POMO|5.04%|6.95%|6.52%|0.75s|
> |LEHD|11.11%|12.73%|12.25%|1.67s|
> |BQ-NCO|10.60%|10.97%|10.89%|3.36s|
> |ELG-POMO (RF)|4.51%|6.46%|6.03%|1.90s|
> |ELG-POMO (PO)|**4.39%**|**6.37%**|**5.94%**|1.90s|
>
> Therefore, these findings verify that PO is also effect in enhancing the generalization ability of RL-based solvers. We will incorporate them into revision.
>
> ---
> **Q1:** Yes, the test set differs from the training set. Training data is generated on-the-fly while test data uses different seeds.
>
> ---
> **Q2:** Fine-tuning is applied after policy convergence with PO, when standard training would not improve performance. Fine-tuning iterations are additional to non-fine-tuning iterations. We will clarify this in revision.
>
> ---
> **Q3:** As PO provides a general algorithmic improvement over REINFORCE variants, integrating it with generalizable RL methods is natural. This includes combining PO with meta-learning like Omni-VRP[r2] or with local policy like ELG[r1].
>
> ---
> We sincerely thank you for your valuable time and insightful comments. We kindly welcome any further questions.
>
> **Reference**
> >[r1] Towards Generalizable Neural Solvers for Vehicle Routing Problems via Ensemble with Transferrable Local Policy. arXiv:2308.14104.
> >
> >[r2] Towards Omni-generalizable Neural Methods for Vehicle Routing Problems. ICML 2023.

---

> > ### Comment · Reviewer_u9d3 · 2025-04-05
> >
> > I appreciate that the authors show now more examples on how their idea can be used in combination with different baseline algorithms, and it improves results in all cases.
> >
> > I don’t follow their response to why one cannot just use the ranks instead of the actual reward values to avoid the issue of diminishing rewards. Most modern evolutionary algorithms use ranks rather than fitness values. Is the issue that it is more challenging to compute the gradient, or because the rank size grows in RL but not in EAs?
> >
> > The authors tune their parameter alpha for every problem but don’t provide guidance on how to set alpha for a new problem in practice. In other cases, I have seen this lead to underwhelming performances when practitioners applied the methods to new applications.
> >
> > The authors compare their algorithm with an additional fine-tuning phase with the algorithm without fine-tuning. At first glance, this doesn’t seem appropriate because the fine-tuning algorithm is allowed to do more learning steps. However, the authors say that the standard algorithm has converged, and so running it longer may not have been beneficial. I am fine with it because the authors agree to make this explicit in the paper.

---

> > > ### Author Response · Authors · 2025-04-07
> > >
> > > We are deeply grateful for your valuable feedbacks and affirmation on the responses. We really appreciate the opportunity to response the further comments, which are presented as follows.
> > >
> > > ---
> > > ### **1. Regarding the comparison with rank-based rewards**
> > >
> > > We appreciate your comment regarding the potential use of rank-based rewards to address the diminishing reward issue. We agree that utilizing ranks instead of actual reward values can indeed circumvent diminishing differences, analogous to *reward shaping* techniques commonly employed in RL community. For instance, if we have five sampled solutions with very similar costs (e.g., 7.773, 7.774, 7.775, 7.776, and 7.777), rank-based methods could reassign their rewards like (5, 4, 3, 2, 1), respectively, to enhance gradient signals in a traditional REINFORCE-based algorithms.
> > >
> > > The proposed PO framework offers two key advantages over standard rank-based reward shaping methods:
> > >
> > > 1) **Integration of probabilistic framework for exploration.** PO naturally stems from entropy regularization within its probabilistic framework to enhance exploration within the solution space. Thereby, PO could *simultaneously address diminishing rewards issues while promoting diverse solution sampling*, as the quantifiable improvements in policy entropy are evidenced in Figures 3(c) and 3(d) of our manuscript.
> > >
> > > 2) **Flexibility without reward engineering.** Rank-based rewards often require careful manual scaling and tailored reward assignments, as gradients in RL are sensitive to reward magnitudes. PO mitigates this issue by relying on flexible and generalizable preference models rather than manual reward engineering.
> > >
> > > ---
> > > ### **2. Our practice of parameter tuning**
> > >
> > > You've identified an important aspect that we should clarify for practical modeling. We appreciate your concern regarding the practical application of our method to new problems, which can be justified from the following aspects.
> > >
> > > + **Methodological views.**
> > > From PO's entropy-regularized objective, the parameter $\alpha$ represents the exploration-exploitation trade-off. Higher $\alpha$ values promote exploration, while lower values emphasize exploitation. In our experiments, we employed a grid search within {0.005, 0.01, 0.05, 0.1, 0.5, 1.0, 2.0} for each problem-model combination.
> > >
> > > + **Empirical views.**
> > > Our empirical findings suggest that PO's performance is influenced by two additional factors:
> > >
> > >   1) *Network architecture's inherent exploration capacity:* Models with built-in exploration mechanisms (e.g., POMO and its variants with multi-start approaches) typically benefit from lower $\alpha$ values to prioritize exploitation but DIMES require more exploration with higher $\alpha$ values. For POMO and its variants on different problems, we observed that routing problems typically perform well with $\alpha$ in the range of 1e-2 to 1e-3, while FFSP benefits from $\alpha$ values between 1.0 and 2.0.
> > >
> > >   2) *Preference model selection:* As PO serves as a flexible framework, different preference models could yield distinct parameterized reward assignments (as shown in Figure 3(b)), necessitating different $\alpha$ calibrations. The Exponential model could be a good candidate when the Bradley-Terry model underperforms on new problems, particularly for challenging problems, before exploring alternatives like Thurstone or Plackett-Luce models (which generalize Bradley-Terry beyond pairwise comparisons). Besides, we also provide detailed analysis regarding different preference models in Appendix E2.
> > >
> > > + **Adapting to new problems.**
> > > For new applications, there are two intuitions for practical extensions:
> > >
> > >   1) *Length-control regularization:* For problems where sampled solutions have varying lengths and shorter solutions with lower costs are preferred, a length-control regularization factor $\frac{1}{|\tau|}$ can be effective, resulting in: $f(\alpha \left[ \frac{1}{|\tau_1|} \log \pi_\theta(\tau_1 | x)- \frac{1}{|\tau_2|} \log \pi_\theta(\tau_2 | x) \right])$.
> > >
> > >   2) *Margin enhancement:* For models with limited capacity, a margin enhancement term $f(\alpha \left[\log \pi_\theta(\tau_1 | x)- \log \pi_\theta(\tau_2 | x) \right] - \gamma)$ can help prioritize better solutions, where $\gamma$ serves as a margin parameter when $f(\cdot)$ is a non-linear function.
> > >
> > > ---
> > >
> > > We sincerely thank you for your insightful comments, and thorough review of our work. Your academic rigor and thoughtful questions have helped us significantly improve both the clarity and quality of our research. We will incorporate your valuable suggestions in our revised manuscript and welcome any further inquiries you may have.

---

### Official Review · Reviewer_SKGq · 2025-03-14

**Overall Recommendation:** 3

**Summary:**

This paper attempts to improve the training paradigm of end-to-end deep learning solvers for combinatorial optimization problems. It introduces Preference Optimization (PO) to alleviate the issue of diminishing advantage signals in the later stages of reinforcement learning (RL) training, thereby preventing models from getting stuck in local optima.

**Claims And Evidence:**

Yes, I believe the evidence is sufficient.

**Essential References Not Discussed:**

n/a

**Experimental Designs Or Analyses:**

The experiment is reliable.

**Methods And Evaluation Criteria:**

Yes

**Other Comments Or Suggestions:**

see above

--------------
Update:
After rebuttal, I raised the score from 2 to 3.

**Other Strengths And Weaknesses:**

Strengths:
- The problem addressed by the authors, namely improving the training paradigm of end-to-end neural solvers, is interesting. Introducing preference optimization into this domain is novel.
- The proposed method is plug-and-play and demonstrates performance improvements across multiple RL-based neural solvers.
Weaknesses:
- The paper only compares RL algorithms enhanced with PO to their original RL counterparts. However, it does not benchmark the improvements against other state-of-the-art (SOTA) approaches from different paradigms, such as:
Supervised learning-based methods (e.g., BQ-NCQ),
Diffusion model-based methods (e.g., Diffusco),
Heuristic learning-based solvers (e.g., NeuroLKH).
- Even if some comparisons do not favor the RL approach, acknowledging the inherent limitations of end-to-end RL methods in this domain and discussing RL’s potential advantages over other paradigms—as well as ways to narrow the performance gap—would add significant value to the paper.
- The problem scale studied in the main text is too small. The authors are encouraged to incorporate the large-scale generalization experiments from the appendix into the main text and compare against relevant SOTA methods to provide a more comprehensive evaluation.

**Questions For Authors:**

see above

**Relation To Broader Scientific Literature:**

See strength

**Theoretical Claims:**

n/a

---

> ### Author Rebuttal · Authors · 2025-04-01
>
> We sincerely thank you for your thorough review and valuable insights. Your constructive feedback has significantly improved our manuscript. We present the detailed responses as follows and hope that could address the concerns.
>
> ---
> ### **1. Broadening comparison with SOTA solvers on large-scale problems and generalization**
>
> We appreciate your suggestion to enhance experimental evaluation. In response, we have included additional comparisons with SOTA solvers LEHD [r1], BQ-NCO [r2], DIFUSCO [r3], NeuroLKH [r4], T2TCO [r5], ELG [r6] on large-scale problems and generalization tasks.
>
> Specifically, recall that our PO approach can serve as algorithmic improvements over baselines, which implies it is applicable to RL-based solvers for large-scale and generalization problems. To validate this claim, we conducted additional experiments with ELG, a neural solver designed for such challenges. Below are the results of optimality gap on TSPLib and CVRPLib with problems scaling up to 1002:
>
> **Results on TSPLib:**
> |Method|(0, 200]|(200, 1002]|Total|Time|Paradigm|
> |-|-|-|-|-|-|
> |LKH3|0.00%|0.00%|0.00%|24s|Heuristic|
> |NeuroLKH|0.00%|0.00%|0.00%|24s|Heuristic+SL|
> |POMO|3.07%|13.35%|7.45%|0.41s|RL|
> |Sym-POMO|2.88%|15.35%|8.29%|0.34s|RL|
> |Omni-POMO|1.74%|7.47%|4.16%|0.34s|RL|
> |Pointerformer|2.31%|11.47%|6.32%|0.24s|RL|
> |LEHD|2.03%|3.12%|2.50%|1.28s|SL|
> |BQ-NCO|1.62%|**2.39%**|**2.22%**|2.85s|SL|
> |DIFUSCO|1.84%|10.83%|5.77%|30.68s|SL|
> |T2TCO|1.87%|9.72%|5.30%|30.82s|SL|
> |ELG-POMO (RF)|1.12%|5.90%|3.08%|0.63s|RL|
> |ELG-POMO(PO)|**1.04%**|5.84%|3.00%|0.63s|RL|
>
> **Results on CVRPLib-Set-X:**
> |Method|(0, 200]|(200, 1000]|Total|Time|Paradigm|
> |-|-|-|-|-|-|
> |LKH3|0.36%|1.18%|1.00%|16m|Heuristic|
> |NeuroLKH|0.47%|1.16%|0.88%|16m|Heuristic+SL|
> |HGS|0.01%|0.13%|0.11%|16m|Heuristic|
> |POMO|5.26%|11.82%|10.37%|0.80s|RL|
> |Sym-POMO|9.99%|27.09%|23.32%|0.87s|RL|
> |Omni-POMO|5.04%|6.95%|6.52%|0.75s|RL|
> |LEHD|11.11%|12.73%|12.25%|1.67s|SL|
> |BQ-NCO|10.60%|10.97%|10.89%|3.36s|SL|
> |ELG-POMO (RF)|4.51%|6.46%|6.03%|1.90s|RL|
> |ELG-POMO (PO)|**4.39%**|**6.37%**|**5.94%**|1.90s|RL|
>
> Consequently, the results demonstrate that the solver trained with our PO method consistently outperforms the original counterpart, verifying that PO enhances generalizability across diverse problem scales. We will incorporate these findings to main text in revision.
>
> ---
> ### **2. Pros and Cons of RL and other paradigms**
>
> We sincerely appreciate your suggestion. We try to expand our discussion on the Pros and Cons of RL over other paradigms as follows.
>
> Generally, RL-based approaches offer flexible training without requiring expert knowledge or high-quality reference solutions. As demonstrated in Table 2, our PO-based solver achieves better performance on FFSP instances where heuristic solvers struggle. We acknowledge that RL may face challenges such as slower convergence rates and longer training times compared to supervised learning (SL) approaches, while SL also induces the intractable problem of acquiring the supervision, e.g., annotations. Note that RL-based methods are free of these limitations, and our PO-based algorithmic improvement can further improve the RL baselines with significantly better results.
>
> Besides, a promising future research direction lies in hybrid approaches combining RL with heuristic methods like MCTS and 2-Opt, as in DIMES. Our experiment results also preliminary verify this point, i.e., the experiment and analysis of training DIMES with PO (included in the Appendix E.3) demonstrate that PO can further enhance such hybrid methods.
>
> ---
> We are deeply grateful for your thorough review and insightful feedback. We sincerely hope that these responses address your concerns, and kindly welcome any further discussions.
>
>
> **References**
> >[r1] [Fu Luo et al., 2023]. Neural Combinatorial Optimization with Heavy Decoder: Toward Large Scale Generalization. NeurIPS 2023.
> >
> >[r2] [Drakulic et al., 2023]. BQ-NCO: Bisimulation Quotienting for Efficient Neural Combinatorial Optimization. NeurIPS 2023.
> >
> >[r3] [Sun et al., 2023]. DIFUSCO: Graph-based Diffusion Solvers for Combinatorial Optimization. NeurIPS 2023.
> >
> >[r4] [Xin et al., 2021]. NeuroLKH: Combining Deep Learning Model with Lin-Kernighan-Helsgaun Heuristic for Solving the Traveling Salesman Problem. NeurIPS 2021.
> >
> >[r5] [Li et al., 2023]. T2T: From Distribution Learning in Training to Gradient Search in Testing for Combinatorial Optimization. NeurIPS 2023.
> >
> >[r6] [Liu et al., 2023]. Towards Generalizable Neural Solvers for Vehicle Routing Problems via Ensemble with Transferrable Local Policy. arXiv:2308.14104.

---

### Official Review · Reviewer_HAxz · 2025-03-14

**Overall Recommendation:** 3

**Summary:**

This paper introduces Preference Optimization (PO), a novel method for solving CO problems like TSP and CVRP.
Key contributions:
1. The authors transform quantitative reward signals into qualitative preference signals, addressing two major challenges in reinforcement learning for COPs:
   - Diminishing reward signals as policy improves
   - Inefficient exploration in vast combinatorial action spaces

2. They reparameterize the reward function in terms of policy and use statistical preference models (like Bradley-Terry) to formulate an entropy-regularized objective that aligns policy directly with preferences.

3. They integrate local search techniques during fine-tuning rather than as post-processing, helping policies escape local optima without adding inference time.


## Update after rebuttal

The authors have partially answered my questions but failed to address all my concerns during the rebuttal. I maintain my rating as weak accept.

**Claims And Evidence:**

Yes.

**Essential References Not Discussed:**

- [Grinsztajn et al., 2023] Winner Takes It All: Training Performant RL Populations for Combinatorial Optimization.
- [Chalumeau et al., 2023] Combinatorial Optimization with Policy Adaptation using Latent Space Search

**Experimental Designs Or Analyses:**

Yes.

**Methods And Evaluation Criteria:**

The proposed method aims at CO problems, for which TSP, CVRP, and FFSP are standard benchmarks.

**Other Comments Or Suggestions:**

None

**Other Strengths And Weaknesses:**

Strength: formulation of the method
Weakness: lacks more recent baselines (e.g. Poppy [Grinsztajn et al., 2023] and COMPASS [Chalumeau et al., 2023])

**Questions For Authors:**

1. Optimizers are mostly scale-invariant, e.g. Adam scales updates to be loss scale invariant. The proposed method can be compared to reward shaping. Would you have ablation studies using reward shaping or reward normalization to understand how PO compares to that?
2. Equation 3: should the entropy term not be inside the expectation over x? (max-entropy framework)

**Relation To Broader Scientific Literature:**

This work brings ideas from preference-based optimization (e.g. in RLHF) to combinatorial optimization.

**Theoretical Claims:**

I have not found any concerns.

---

> ### Author Rebuttal · Authors · 2025-04-01
>
> We sincerely thank you for your valuable feedback and constructive comments. Your insightful suggestions are invaluable for improving our work. Below, we address your concerns in detail.
>
> ---
> ### **1. Including comparison with Poppy and COMPASS**
>
> Thank you for providing the related references, i.e., COMPASS [r1] and Poppy [r2]. We discuss them from the following aspects, and will incorporate them into revision.
>
> *Methodological Comparison.* In general, both them and our work focus on RL for COPs, while the major difference is that the provided works focus on the framework design for enhancing the diversity of learned policies, and our proposed PO algorithm concentrates on algorithmic improvements on the quality of policies and efficiency of learning, which can serve as a plug-and-play objective to the baselines with REINFORCE-based objective.
>
> *Experimental Validation.* To validate our plug-and-play PO objective on these baselines, we try to reproduce the baselines with the official open-source codes and adapted our objective for further algorithmic improvement, i.e., replacing the policy gradient-based training with PO-based training.
>
> Due to computational constraints (each experiment requires approximately 50 hours on a single 80GB-A800 GPU for 1e5 training steps, with complete training requiring over 60 days), we provide the preliminary comparison results of current training records, including the models trained from scratch with original baselines and PO-based improvement:
>
> *Results on TSP-100:*
> |Step|10k|20k|30k|40k|50k|60k|70k|80k|90k|100k|
> |:-:|:-:|:-:|:-:|:-:|:-:|:-:|:-:|:-:|:-:|:-:|
> |COMPASS(RF)|75.29%|29.69%|32.43%|51.36%|34.44%|39.23%|27.14%|17.22%|14.02%|10.93%|
> |COMPASS(PO)|9.51%|7.15%|5.89%|4.95%|4.50%|4.32%|4.14%|3.90%|3.76%|3.61%|
> |Poppy(RF)|4.57%|3.85%|3.17%|2.83%|2.59%|2.39%|2.39%|2.24%|2.05%|1.95%|
> |Poppy(PO)|2.28%|1.56%|1.23%|1.03%|0.92%|0.78%|0.71%|0.69%|0.66%|0.63%|
>
> *Results on CVRP-100:*
> |Step|10k|20k|30k|40k|50k|60k|70k|80k|90k|100k|
> |:-:|:-:|:-:|:-:|:-:|:-:|:-:|:-:|:-:|:-:|:-:|
> |COMPASS(RF)|120.82%|58.17%|43.11%|20.38%|14.80%|12.78%|11.46%|10.84%|10.40%|9.79%|
> |COMPASS(PO)|18.84%|10.26%|8.99%|8.10%|7.47%|7.19%|6.91%|6.53%|6.04%|6.50%|
> |Poppy(RF)|14.15%|11.13%|9.42%|8.52%|8.63%|7.16%|6.94%|6.15%|5.91%|5.42%|
> |Poppy(PO)|6.25%|4.54%|3.77%|3.43%|3.11%|2.95%|2.78%|2.65%|2.53%|2.43%|
>
> The results above demonstrate that 1) PO significantly ensures lower optimality gap at the same iteration number; 2) PO ensures much faster convergence speed for the same gap, and higher stability during optimization. Moreover, these results also validate that our proposed algorithmic improvement method is consistently effective in various RL-based baselines.
>
> ---
> ### **2. Regarding comparison to reward shaping**
>
> We fully agree with you about the importance of comparing PO with reward shaping methods, which is just right considered in experiment section. Specifically, the original Pointerformer implementation already incorporated reward normalization techniques:
> $\nabla_{\theta} J(\theta) \approx \frac{1}{B \times N} \sum_{i=1}^{B} \sum_{j=1}^{N}\left(\frac{R(\tau_i^j)-\mu(\tau_i)}{\sigma(\tau_i)}\right)\nabla_{\theta}\log p_{\theta}\left(\tau_i^j \mid s\right)$.
> As shown in Table 1 and Figure 2(c) of our manuscript, the results consistently show that PO outperforms its REINFORCE variant with reward shaping. Figure 3(b) also illustrates how PO re-distributes the advantage values.
>
> Besides, from theoretical view, the fundamental difference between PO and reward shaping lies in the fact that **PO is derived from an entropy-regularized objective and is invariant to reward shaping since that would not change the preference relationship between solutions**, making it more suitable for exploring large discrete spaces, while reward shaping functions more as a technique to stabilize the training of RL policies. In current manuscript, the analysis is already included in Appendix E.2, i.e., the different preference models (e.g., Bradley-Terry, Thurstone, Exponential) which is used to control the advantage values assignment like reward shaping.
>
> ---
> ### **3. Correction of Equation 3**
>
> We sincerely thank you for your careful examination and valuable suggestion. We will revise Equation 3 to the more precise form:
> $\max_{\pi_{\theta}} E_{x \sim \mathcal{D}, \tau \sim \pi_{\theta}(\tau|x)}\left[ r(x,\tau) + \alpha \mathcal{H}\left(\pi_{\theta}(\tau|x)\right)\right]$.
>
> ---
> We are deeply grateful for your thoughtful feedbacks, and sincerely hope that these responses address your concerns. We welcome any further discussions or suggesstions.
>
> ### **Reference**
> >[r1] [Grinsztajn et al., 2023] Winner Takes It All: Training Performant RL Populations for Combinatorial Optimization. NeurIPS 2023
> >
> >[r2] [Chalumeau et al., 2023] COMPASS: Cooperative Multi-Agent Policy Search for Combinatorial Optimization. NeurIPS 2023

---

### Decision · Program_Chairs · 2025-05-01

**Decision:**

Accept (poster)

**Comment:**

This study explores the concept of qualitative preference optimization to address important challenges in applying reinforcement learning to combinatorial optimization. All reviewers recognize the novelty of the preference optimization framework and highlight the substantial improvements it offers compared to current state-of-the-art methods. For these reasons, I recommend that the paper be accepted.

Given that the authors have engaged in productive discussions with the reviewers, it would be beneficial to incorporate these insights into the final version of the paper. Key suggestions include clarifying the rationale behind the framework, explaining the choice of parameters, and discussing parameter tuning. Additionally, the authors should include detailed comparisons with other baselines, such as Compass and Poppy, and conduct more experiments using SOTA solvers to highlight the framework's flexibility.